# Challenges in Non-Polymeric Crystal Structure Prediction: Why a Geometric, Permutation-Invariant Loss is Needed

**Emmanuel Jehanno**                                     *emmanuel.jehanno@inria.fr*
*Univ. Grenoble Alpes, Inria, CNRS, Grenoble INP, LJK*
*38000 Grenoble, France*

**Romain Menegaux**                                     *romain.menegaux@gmail.com*
*work done while at Univ. Grenoble Alpes and Inria*

**Julien Mairal**                                       *julien.mairal@inria.fr*
*Univ. Grenoble Alpes, Inria, CNRS, Grenoble INP, LJK*
*38000 Grenoble, France*

**Sergei Grudinin**                                     *sergei.grudinin@univ-grenoble-alpes.fr*
*Univ. Grenoble Alpes, CNRS, Grenoble INP, LJK*
*38000 Grenoble, France*

**Reviewed on OpenReview:** *https://openreview.net/forum?id=MsIi78JXXZ*

## Abstract

Crystalline structure prediction is an essential prerequisite for designing materials with targeted properties. Yet, it is still an open challenge in materials design and drug discovery. Despite recent advances in computational materials science, accurately predicting three-dimensional non-polymeric crystal structures remains elusive. In this work, we focus on the molecular assembly problem, where a set $\mathcal{S}$ of identical rigid molecules is packed to form a crystalline structure. Such a simplified formulation provides a useful approximation to the actual problem. However, while recent state-of-the-art methods have increasingly adopted sophisticated techniques, the underlying learning objective remains ill-posed. We propose a better formulation that introduces a loss function capturing key geometric molecular properties while ensuring permutation invariance over $\mathcal{S}$. Remarkably, we demonstrate that within this framework, a simple regression model already outperforms prior approaches, including flow matching techniques, on the COD-Cluster17 benchmark, a curated non-polymeric subset of the Crystallography Open Database (COD). *Our code is available at https://github.com/inria-thoth/SinkFast.*

## 1 Introduction

On the one hand, for *material property prediction*, advances in graph neural networks and transformers have significantly improved the understanding of molecular structures (Joshi et al., 2023; Lin et al., 2023; Choudhary & DeCost, 2021), linking their three-dimensional (3D) geometry to physical and chemical properties. Particular attention has been paid to SE(3)-equivariant representations, which present higher expressivity by preserving geometric symmetries (Schütt et al., 2021). These methods have been adapted to crystalline structures, with their inherent challenges of infinite periodicity and rich symmetry patterns (Yan et al., 2024a). Yan et al. (2022; 2024a); Ito et al. (2025) yield state-of-the-art performance in property prediction of crystalline structures thanks to physically grounded methods, reflecting the need to integrate physics knowledge in models. On the other hand, for *material design*, generative models such as diffusion models (Song et al., 2021) and flow matching methods (Liu et al., 2023) have greatly enhanced the capacity to generate valid and diverse molecular and material structures (Watson et al., 2023). This work aims to

combine these two aspects for the task of *molecular assembly prediction*, where a finite set of identical rigid molecules is packed into a crystalline structure.

A fundamental step in designing a material with specific properties is to know its crystallization pattern. As represented in Figure 1, a crystal is conventionally described by a unit cell, the smallest volume that contains all the structural and symmetry information necessary to generate the whole crystal by translation. This three-dimensional infinitely periodic shape largely determines the physical and chemical properties of the resulting material. This shape can be predicted either by regression (Liang et al., 2020; Cao et al., 2024) or by flow matching/diffusion methods that allow for probabilistic answers (Merchant et al., 2023; Xie et al., 2022; Luo et al., 2025; Pakornchote et al., 2024; Jiao et al., 2023).

Most of the previous methods model atoms in the unit cell individually. While such an approach works well (Miller et al., 2024) for simple crystals of atomic point clouds from the Materials Project (Jain et al., 2013), the performance degrades on more complex molecular materials with symmetries other than translations. These contain internal *point-group* symmetries within the unit cell. An asymmetric unit (ASU) is defined as an elementary pattern of the unit cell, irreducible under the symmetry group transformations. A unit cell can be composed of multiple ASUs and an example is shown in Figure 1A. As this basic structure maintains a fixed internal structure, generating the crystal by directly predicting the ASU position, orientation, and symmetry operations in the world frame significantly reduces the dimensionality of the task, compared to moving each atom individually. In this setting, the goal of the *molecular assembly prediction* problem can be formulated as follows: given an elementary structure – an ASU –, predict its local crystalline structure, or in other words, how it packs in space.

**Contributions** In this work we show the need of integration of domain-specific physics knowledge in the training scheme of models and the challenges that constitute the task of material generation. Our contributions can be summarised as follows:

1. **Physics grounded loss.** We show that a domain-specific rigid-body, model agnostic loss, grounded in physical principles, leads to improved prediction of crystalline structures.

2. **Permutation-invariant loss.** We propose an effective differentiable *soft matching* objective that is invariant to global geometric transformations and to the order permutation of repeated molecular units.

3. **Remaining challenges.** While the proposed domain-driven learning objective enables us to outperform prior approaches with a simple regression model, we also witness the challenges that remain to be tackled to reach real-world applicability.

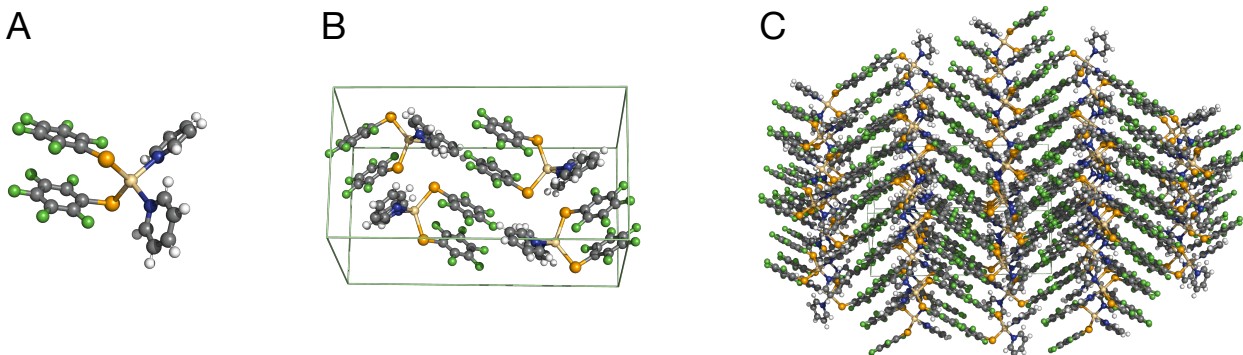

Figure 1: A crystalline material at three different scales. From left to right: (A) The asymmetric subunit (ASU). (B) The unit cell with mirror images of the ASU. (C) The unit cell is repeated periodically in all three directions. Illustrations correspond to the COD-4316210 crystal structure from Crystallographic Open Database (Gražulis et al., 2009).

## 2 Related Works

**Datasets.**   The fast-moving field of materials science has seen significant advances in recent years, largely driven by the release of large-scale open-source datasets. Many of the works discussed here rely on the QM9 database (Ramakrishnan et al., 2014), the Materials Project (Jain et al., 2013) and JARVIS (Choudhary et al., 2020). While these datasets contain only polymeric and inorganic crystals, the Crystallography Open Database (Gražulis et al., 2009) is the main open-source dataset for experimentally determined non-polymeric organic crystals, composed of 507k structures. OMC25 (Gharakhanyan et al., 2025), an even larger database composed of 25 million crystals, was recently released and can be used to define new bigger benchmarks. However, OMC25 consists of simulated zero-temperature structures that may not corresponds to those observed in X-ray experiments, unlike the COD data. Thus in this work we focus on COD-Cluster17 (Liu et al., 2024b), a dataset sanitized from COD.

### 2.1 Physics informed GNN for property prediction

Graph Neural Networks (GNNs) with message passing (Kipf & Welling, 2017; Rampášek et al., 2022) and transformer-based architectures (Ying et al., 2021; Menegaux et al., 2023) have been widely applied to molecular property prediction. Initially adapted from 2D molecular representations, GNNs have been extended to crystalline materials. Notable models include CGCNN (Xie & Grossman, 2018), MEGNet (Chen et al., 2019), and GATGNN (Louis et al., 2020), which pioneered the application of GNNs to materials property prediction. To better capture the geometric and physical properties of materials, geometry-aware GNNs have been developed (Duval et al., 2023). Physically grounded models such as ALIGNN (Choudhary & DeCost, 2021), Matformer (Yan et al., 2022), PotNet (Lin et al., 2023) and ComFormer (Yan et al., 2024a) achieve state-of-the-art results on the Materials Project dataset, demonstrating the importance of incorporating materials science knowledge into predictive models. Concurrently, SE(3)-equivariant methods, known for their expressivity, have emerged with models such as SchNet (Schütt et al., 2017), PaiNN (Schütt et al., 2021), SEGNN (Brandstetter et al., 2022), SphereNet (Liu et al., 2022), NequIP (Batzner et al., 2022) and Equiformer (Liao & Smidt, 2023). Such models can be used as backbones to provide powerful molecular representations that are essential for generative models in crystal structure prediction (Guo et al., 2025).

### 2.2 Geometric representations in computer vision

In computer vision, permutation-invariant loss functions have been used and developed in multiple object detection and segmentation (Carion et al., 2020) and multi-object tracking (Xu et al., 2020). Locatello et al. (2020) and Kori et al. (2024) learn a binding scheme for assigning objects to slots in object property prediction and unsupervised instance discovery. In the point cloud registration domain, Wang & Solomon (2019) have studied rigid alignment of point clouds as well as prediction to target assignment. However, they decorrelate $\mathbb{R}^3$ and SO(3) in the loss and reassign predictions to target only when correspondence is unknown. Pais et al. (2019) study the registration of 3D scans and learn the rigid alignment using different distances. Park et al. (2020) use Procrustes-alignment of 3D shapes to learn a regression problem of predicting 3D positions of a deformable object from 2D frame observations. Here, we transfer this powerful and effective knowledge to material science in order to provide a robust and physics grounded training scheme.

### 2.3 Generative models in materials science

Generating the 3D stable configuration of a single molecule is essential for materials discovery. Datasets such as GEOM-Drugs (Axelrod & Gomez-Bombarelli, 2022) and OMol25 (Levine et al., 2025) are tailored for this task. The OMol25 dataset includes evaluations based on linear sum assignment for assessing optimal conformers, guided by machine learning interatomic potentials (Smith et al., 2017) and Density Functional Theory (DFT) (Kohn & Sham, 1965). Generative approaches include flow matching models and SE(3)-equivariant generative models such as those by Cornet et al. (2024) and Song et al. (2023).

Historically, the problem of computational material design—predicting the global arrangement of such molecules in space— has been extensively studied through the lens of Crystal Structure Prediction (CSP) challenge. At first, CSP has relied on computationally expensive iterative energy assessment of predicted

structures with first-principles calculations based on the density functional theory (DFT) (Kohn & Sham, 1965; Kresse & Furthmüller, 1996; Pickard & Needs, 2011), including techniques by Wang et al. (2021); Glass et al. (2006), where atoms are iteratively replaced by chemically similar ones and validated with DFT calculations. Recently, machine learning has accelerated this process (Schmidt et al., 2022; Merchant et al., 2023). For example, Genarris 3.0 (Yang et al., 2025) combines a physically-constrained optimization with machine learning-based interatomic potentials.

For simple crystals from the Materials Project (Jain et al., 2013), their 3D infinitely periodic structures can now be directly predicted (Liang et al., 2020; Cao et al., 2024). These methods are further enhanced by diffusion models (Merchant et al., 2023; Xie et al., 2022; Pakornchote et al., 2024; Jiao et al., 2023; Levy et al., 2025) and flow-matching approaches (Miller et al., 2024; Luo et al., 2025; Nam et al., 2025). Inspired by their success in other domains, Large Language Models have been adapted to CSP, as seen in CrystalLLM (Antunes et al., 2024) and models that integrate SE(3) equivariance and periodic boundary conditions (Yan et al., 2024b).

For more complex molecular structures, rigid-body generative models are extensively explored in protein design and structure prediction, as in AlphaFold2 (Jumper et al., 2021), FrameDiff (Yim et al., 2023b), and FrameFlow (Yim et al., 2023a). Closer to molecular crystals, studies now focus on assembly prediction, where a finite *cluster* of molecules is packed into a pattern that is able to replicate a crystal structure—we thoroughly define it in Section 3. For example, Liu et al. (2024b) propose atom-wise equivariant flow matching, while Guo et al. (2025) introduce a rigid body flow matching model for molecular cluster packing prediction. Here, we propose a physically-grounded training scheme with a deep learning prediction model and explore the remaining challenges of the cluster packing prediction task.

## 3 Problem setting

**Problem formulation**  A non-polymeric crystal is a solid material in which molecules are arranged in a highly ordered pattern—the unit cell—repeating in the three spatial dimensions (3D). The asymmetric units ASU that constitute it are molecules that are identical objects in 3D. The unit cell is then defined by a finite number of symmetry operators applied to the ASU. The pinnacle of crystalline structure prediction is to compute the infinite 3D structure of a material given its substituent chemical compounds. To solve this challenging task, one can make a number of approximations and hypotheses. The molecular assembly subproblem is a simplification of the original problem, where a finite set $\mathcal{S}$ of identical rigid molecules is packed together from a state $\mathcal{S}_{\text{initial}}$ of randomly positioned molecules in space, into a state $\mathcal{S}_{\text{final}}$ which forms a pattern that can be then replicated in space into a crystal. Our goal is thus to predict rigid spatial transformations $\mathcal{T}_i$ for each molecule $i$ that reconstruct the $\mathcal{S}_{\text{final}}$ set from the $\mathcal{S}_{\text{initial}}$ set. We propose an efficient and model-agnostic way to guide any machine learning model with physical knowledge of the task.

**Dataset**  In this work we use the COD-Cluster17 assembly dataset introduced by Liu et al. (2024b), specifically constructed for the task of non-polymeric crystal structure prediction. To the best of our knowledge, this is currently the only available curated benchmark for this task. This dataset contains 111k assemblies and is a simplified, sanitized version of the 507k crystals from the real world Crystallography Open Database (COD) (Gražulis et al., 2009). The procedure to build the dataset is detailed in (Liu et al., 2024b) and can be summarized as follows. First, crystals are extracted from the COD if: (1) their asymmetric unit contains only one molecule; (2) they do not present disordered atoms (cases where some atoms do not occupy unique and uniquely attributed positions); (3) they are non-polymeric. Then, the dataset is built by computing for each filtered crystal the ground-truth supercell of an arbitrary asymmetric unit–referred to as the *central molecule*–, which is the aggregation of 27 unit cells into a parallelepiped centered on the unit cell of the asymmetric unit of interest. An example of a supercell is given in Figure 1C. The authors of COD-Cluster17 then extracted the central molecule's 16 nearest neighbors using a cutoff in Euclidean space within this supercell. This procedure outputs the *final positions* set consisting of each atom Cartesian coordinates. Then, a random rigid-body transformation is applied to the atomic positions of each molecule, which results in the *initial positions* set. The task for the COD-Cluster17 benchmark is then originally a point cloud packing matching task of predicting all atoms final absolute positions, provided the known

correspondence with the initial positions. This task has also been formulated as a rigid-body packing matching by Guo et al. (2025) as molecular integrity is preserved in both $\mathcal{S}_{\text{initial}}$ and $\mathcal{S}_{\text{final}}$ sets.

However, the exact mapping enforcing specific index correspondences between the assembly atoms or molecules is unrealistic as the mapping is arbitrary. Indeed, all ASUs in a crystal are geometrically, physically and chemically equivalent. Thanks to the filtering procedure of the COD-Cluster17 dataset construction, as detailed above, the 17 molecules in $\mathcal{S}_{\text{initial}}$ and $\mathcal{S}_{\text{final}}$ sets are thus also equivalent. Then, there is no specific reason why $i$ in $\mathcal{S}_{\text{initial}}$ must be associated with $i$ but not a different index $j$ in $\mathcal{S}_{\text{final}}$. One of our contributions is thus to propose a more reasonable task of packing molecules without enforcing index correspondences, preserving the invariance to permutations of the set $\mathcal{S}_{\text{final}}$ of 17 molecules. It is important to note that despite the COD-Cluster17 benchmark is available in two distinct versions – with and without molecular inversion – Guo et al. (2025) focus exclusively on the version without inversions for simplicity. While this assumption ensures that molecules are identical under rigid transformations, we argue in Appendix C.1 that our method can also be efficiently adapted to the inversion dataset.

**Rigid body description**   We aim to predict the positions of rigid molecules in 3D, which can not be described by single position vectors as for atoms. Instead, we represent the position of a rigid molecule as a rigid spatial transformation operator $\mathcal{T} = (\vec{r}, \mathbf{q})$ composed of a 3D translation $\vec{r} \in \mathbb{R}^3$ and a 3D rigid rotation quaternion $\mathbf{q} = [s, \vec{q}] \in \text{SO}(3)$, where $s$ is its scalar part and $\vec{q}$ is its vector part. See Appendix A for details.

## 4   Methods

We will now refer to the global reference frame as an arbitrarily chosen coordinate system used to represent the positions and orientations of the $M$ molecules in the set $\mathcal{S}$. In contrast, we define local frames as those attached to the center of mass of each individual molecule, which rigidly move with them. These local frames are initialized using the principal components of each molecule's inertia tensor. While this initialization may not be unique, the specific choice does not affect the relative transformations between local frames, which are the quantities actually used in the model computations.

### 4.1   Metrics

**Packing matching**   Current crystal structure prediction methods (Guo et al., 2025; Liu et al., 2024b) typically use the *Packing Matching* (PM) score as an assessment metric. It is defined for atoms positions $\vec{x}$ as follows,

$$\text{PM}_{\text{atom}}^2 = \frac{1}{N^2} \sum_{i=1}^{N} \sum_{j=1}^{N} \left( \|\vec{x}_i^{\text{pred}} - \vec{x}_j^{\text{pred}}\| - \|\vec{x}_i^{\text{gt}} - \vec{x}_j^{\text{gt}}\| \right)^2, \tag{1}$$

where $N$ is the number of atoms in the assembly, $\vec{x}_i^{\text{pred}}$ (resp. $\vec{x}_j^{\text{gt}}$) is the position vector of atom $i$ (resp. $j$) in the predicted (resp. ground-truth) assembly. PM quantifies how well the pairwise distances between the atoms are predicted, and is invariant to global rotations and translations. Also commonly used, the $\text{PM}_{\text{center}}$ metric, is defined as follows,

$$\text{PM}_{\text{center}}^2 = \frac{1}{M^2} \sum_{i=1}^{M} \sum_{j=1}^{M} \left( \|\vec{c}_i^{\text{pred}} - \vec{c}_j^{\text{pred}}\| - \|\vec{c}_i^{\text{gt}} - \vec{c}_j^{\text{gt}}\| \right)^2, \tag{2}$$

where $M$ is the number of molecules in the assembly, $\vec{c}_i^{\text{pred}}$ (resp. $\vec{c}_j^{\text{gt}}$) is the position of $i$th molecule's (resp. $j$th) center of mass in the predicted (resp. ground-truth) assembly. It evaluates the quality of the molecule positions regardless of their orientations.

**Root mean square displacement**   RMSD is another metric common in chemistry, structural biology, physics, and materials science. RMSD performs direct comparisons of atom positions, which requires representing them in a common frame:

$$\text{RMSD}_{\text{atom}}^2 = \frac{1}{N} \sum_{i \in N} \|\vec{x}_i^{\text{pred}} - \vec{x}_i^{\text{gt}}\|^2. \tag{3}$$

Note that Appendix B proves the relation $\text{PM}_{\text{atom}} \leq \sqrt{2}\text{RMSD}_{\text{atom}}$ showing the correlation between both metrics even though PM compares relative positions, whereas RMSD relies on absolute ones. This relation shows that PM score is a good proxy for the RMSD assessment. In particular: "*as long as reported PM is greater than 2 times square root of 2 ångströms, RMSD is greater than 2 ångströms.*".

$\mathcal{S}$-**Permutation invariant metric**    The molecular assembly task as defined in this paper aims at matching a set $\mathcal{S}_{\text{initial}}$ of $M$ equivalent initial molecules to a set $\mathcal{S}_{\text{final}}$ of $M$ final ones. As these molecules are identical, positioning one at a given place or another is strictly equivalent physically. A proper metric should thus reflect this $\mathcal{S}$-permutation invariant property, which we now present.

We represent molecules $i$ from the predicted assembly and $j$ from the ground truth assembly by their rigid-body positions $\mathcal{T}_{\text{pred}}^i$ and $\mathcal{T}_{\text{gt}}^j$ in the global reference frame, from which we can reconstruct atoms positions $\vec{x}_i^{\text{pred}}$ and $\vec{x}_j^{\text{gt}}$ to compute the PM scores. We consider the cost matrix $\mathcal{C}^{\mathcal{L}}$ of any metric $\mathcal{L}$ such as $\text{PM}_{\text{atom}}$ or $\text{PM}_{\text{center}}$, such that $\mathcal{C}_{ij}^{\mathcal{L}}$ is the cost of assigning molecule $i$ from the ground truth assembly with the molecule $j$ in the predicted assembly. This cost matrix is computed as follows:

$$\forall \{i,j\} \in [\![1,M]\!]^2, \quad \mathcal{C}_{ij}^{\mathcal{L}} = \mathcal{L}\left(\mathcal{T}_{\text{pred}}^i, \mathcal{T}_{\text{gt}}^j\right). \tag{4}$$

The goal is then to find a complete assignment of molecules in the predicted assembly with molecules in the ground truth assembly, which minimizes the metric $\mathcal{L}$ over all $\mathcal{S}$-permutations. This minimizer is denoted $\mathcal{L}^*$. Formally it is defined by the linear sum assignment problem. Let $P$ be a boolean pairing matrix in which $P_{ij} = 1$ if and only if molecule $i$ from ground truth assembly is mapped with molecule $j$ in the predicted assembly:

$$\mathcal{L}^* := \min_P \sum_{ij} C_{ij}^{\mathcal{L}}.P_{ij} \qquad \text{with } P_{ij} \in \{0,1\} \quad \text{s.t. } P \cdot \mathbf{1} = P^\top \cdot \mathbf{1} = 1. \tag{5}$$

In practice we use scipy's linear sum assignment method to compute this exact minimizer $\mathcal{L}^*$ of $\mathcal{L}$.

## 4.2    Physically grounded losses

While the previous paragraph introduces useful permutation invariant atom-wise metrics, well suited for evaluating atomistic predictions, we now turn to defining trainable objectives that are better adapted to the rigid-body formulation of the task. Concretely, we propose two rigid-body loss functions: $\mathcal{L}_{\text{RMSD}}$, which operates on absolute positions of molecules, and $\mathcal{L}_{\text{Geom}}$, which extends this formulation to relative molecular positions. We will further show how these objectives can be made $\mathcal{S}$-permutation invariant through differentiable optimal assignment in section 4.3.

We now consider rigid body predicted (resp. ground-truth) positions in the global reference frame as $\mathcal{T}_{\text{pred}} = (\vec{r}_{\text{pred}}, \mathbf{q}_{\text{pred}})$ (resp. $\mathcal{T}_{\text{gt}} = (\vec{r}_{\text{gt}}, \mathbf{q}_{\text{gt}})$). The loss currently used in the literature decouples $\mathbb{R}^3$ and SO(3) spaces as:

$$\mathcal{L}_{\mathbb{R}^3}(\mathcal{T}_{\text{pred}}, \mathcal{T}_{\text{gt}}) = \|\vec{r}_{\text{pred}} - \vec{r}_{\text{gt}}\|^2 \qquad\qquad \mathcal{L}_{\text{SO(3)}}(\mathcal{T}_{\text{pred}}, \mathcal{T}_{\text{gt}}) = \|\mathbf{q}_{\text{pred}} - \mathbf{q}_{\text{gt}}\|^2, \tag{6}$$

and then one can combine them with a tuned hyperparameter $\alpha$ as

$$\mathcal{L}_{\text{ML}} = \mathcal{L}_{\mathbb{R}^3} + \alpha \mathcal{L}_{\text{SO(3)}}. \tag{7}$$

The $\alpha$ parameter has to be adjusted to the task one is trying to solve. It has to balance the weight of unbounded distance in $\mathbb{R}^3$ to the bounded distance in SO(3). As different samples in the dataset may have very different geometries, with inter-molecular distances spanning orders of magnitudes, having a single parameter is suboptimal. Finally, as the space of rigid transformations SE(3) is not a *direct product* of $\mathbb{R}^3$ and SO(3), this loss has no physical or geometrical meaning.

**Rigid RMSD loss**    Popov & Grudinin (2014) introduced a more suitable rigid-body transformation loss that is *strictly equivalent* to the $\text{RMSD}_{\text{atom}}^2$ metric in eq. 3. It is defined for a rigid transformation $\mathcal{T} = (\vec{r}, \mathbf{q})$, with quaternion $\mathbf{q} = (s, \vec{q})$ composed of a scalar $s$ and a vector part $\vec{q}$, as follows.

$$\text{RMSD}^2(\mathcal{T}, \mathfrak{I}) = \frac{4}{N} \vec{q}^{\top} \mathfrak{I} \vec{q} + \vec{r}^{\,2}, \tag{8}$$

where $\mathfrak{I}$ is an inertia tensor of the rigid body computed in its center-of-mass local frame (see Appendix A for the definition). One can notice that in this frame the two $\text{RMSD}^2$ contributions, rotation and translation, are additive. The inertia tensor naturally provides a weight between the rotation and the translation contributions. However, the cross-terms appear in the equation if we change the reference frame as detailed in Appendix A. Thus, given two spatial transformations $\mathcal{T}_{\text{pred}}$ and $\mathcal{T}_{\text{gt}}$, of the same rigid body with the inertia tensor $\mathfrak{I}$, we can *naturally* define the physically-grounded RMSD loss without additional hyperparameters as

$$\mathcal{L}_{\text{RMSD}}(\mathcal{T}_{\text{pred}}, \mathcal{T}_{\text{gt}}) = \text{RMSD}^2(\mathcal{T}_{\text{gt}} \circ \mathcal{T}_{\text{pred}}^{-1}, \mathfrak{I}). \tag{9}$$

We will use this RMSD loss as default during training and test to compare absolute positions in the predicted assembly of molecules with the ground truth.

**Geometric loss** Regarding the task of molecular assembly prediction, we aim to define a loss that better reflects the *relative packing* of molecules and not memorizing their absolute positions. Let us consider two rigid molecules in an assembly $\mathcal{S}$ consisting of $M$ molecules, $i, j \in M$. Let us assume we have predicted a packing $\mathcal{S}_{\text{pred}}$ of these molecules resulting in individual global spatial transformations (or positions) $\mathcal{T}_{i,\text{pred}}$ and $\mathcal{T}_{j,\text{pred}}$. We want to compare these transformations to the corresponding ground-truth ones $\mathcal{T}_{i,\text{gt}}$ and $\mathcal{T}_{j,\text{gt}}$ from the packing $\mathcal{S}_{\text{gt}}$. We can define the assembly transformation-invariant PM metric for these molecules similar to the one in eq. 1. However, this computation requires to first compute positions of all corresponding atoms. Instead, we propose a more elegant rigid-body RMSD-based solution presented in Figure 2. Concretely, we compute the $\text{RMSD}^2$ metric between the $i$th molecules in the superposed local frames of the $j$th molecules, as follows,

$$\mathcal{L}_{\text{RMSD}}(\mathcal{T}_{j,\text{pred}}^{-1} \circ \mathcal{T}_{i,\text{pred}}, \mathcal{T}_{j,\text{gt}}^{-1} \circ \mathcal{T}_{i,\text{gt}}) = \text{RMSD}^2(\mathcal{T}_{j,\text{gt}}^{-1} \circ \mathcal{T}_{i,\text{gt}} \circ \mathcal{T}_{i,\text{pred}}^{-1} \circ \mathcal{T}_{j,\text{pred}}, \mathfrak{I}). \tag{10}$$

We can then extend the above expression to the comparison of $M - 1$ molecules to a *reference* one. Without loss of generality, we can assume it is the $M$th molecule and define the *geometric loss* $\mathcal{L}_{\text{Geom}}$ as follows:

$$\mathcal{L}_{\text{Geom}}(\mathcal{T}_{\text{pred}}, \mathcal{T}_{\text{gt}}) = \frac{1}{M-1} \sum_{i=1}^{M-1} \mathcal{L}_{\text{RMSD}}(\mathcal{T}_{M,\text{pred}}^{-1} \circ \mathcal{T}_{i,\text{pred}}, \mathcal{T}_{M,\text{gt}}^{-1} \circ \mathcal{T}_{i,\text{gt}}). \tag{11}$$

In practice in COD-Cluster17, as detailed in section 3, the way the dataset is constructed defines a reference molecule around which we extract the 16 nearest neighbors. We show some illustration of the proposed losses in Appendix A.7.

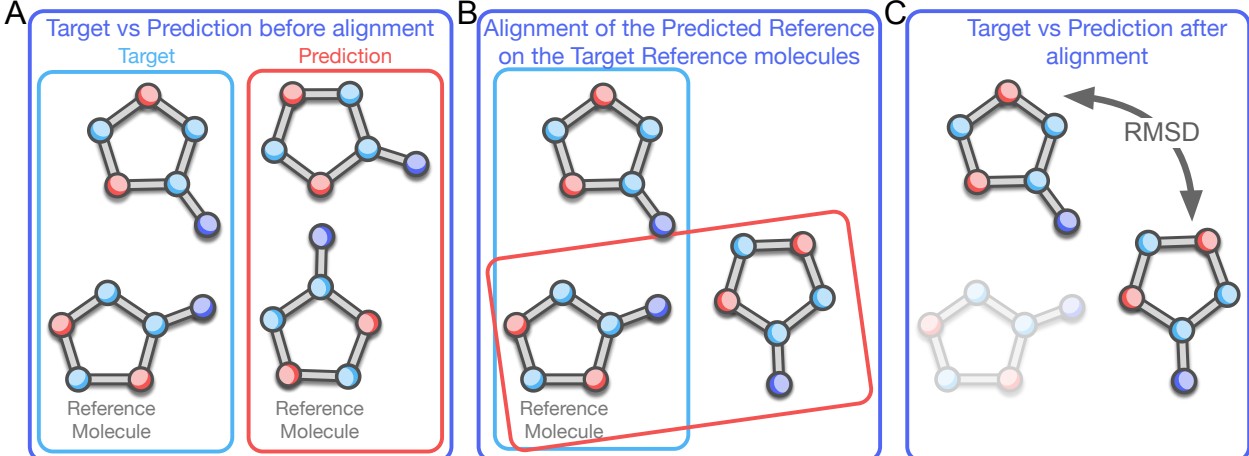

Figure 2: A schematic illustration of our geometric loss alignment and similarity measure computation. *A:* Predicted and target couples of molecules with local frames before alignment. *B:* The reference molecule from the predicted packing is aligned on the one from the target packing. *C:* Predicted and target molecules after aligning both reference molecules on each other. The similarity measure is then computed as the $\text{RMSD}^2$ between the non-reference molecules.

### 4.3 Differentiable optimal assignment

As metrics are computed with invariance to $\mathcal{S}$-permutations, it is essential to also train models with permutation invariant losses. However, the linear sum assignment problem 5 is not differentiable and results in training instabilities, as our preliminary experiments demonstrated. We thus use during training the differentiable version of it provided by the Sinkhorn algorithm with the boolean pairing $P$ matrix being relaxed as $\forall\{i,j\}, 0 < P_{ij} < 1$. The problem is then defined for any training loss $\mathcal{L}$, like $\mathcal{L}_{\mathrm{ML}}, \mathcal{L}_{\mathrm{RMSD}}$ or $\mathcal{L}_{\mathrm{Geom}}$, such that:

$$
\begin{aligned}
&\mathcal{L}_{\mathrm{train}}^* = \min_P \langle P, \mathcal{C}^{\mathcal{L}} \rangle_F + \mathrm{reg} \cdot \Omega(P) \\
&\text{with } P_{ij} \in [0,1] \quad \text{s.t. } P \cdot \mathbf{1} = P^\top \cdot \mathbf{1} = 1 \text{ and } P \geq 0 \\
&\text{with } \Omega(P) = \sum_{ij} P_{ij}\log(P_{ij})
\end{aligned}
\tag{12}
$$

An implementation of this algorithm as defined in Cuturi (2013) can be found in Python Optimal Transport library (Flamary et al., 2021). This approach provides a feedback to the model with multiple possible assignments weighted by $P$, which behaves like a *probability map*.

## 5 Results

### 5.1 Experimental setup

**Dataset**  We evaluate the performance of our approach on the COD-Cluster17 benchmark introduced in Liu et al. (2024b). The dataset and the task are detailed in section 3. It contains 111k assemblies and is a simplified, sanitized version of the 507k crystals from the real world Crystallography Open Database. Previous methods also benchmark on a subset of *5k* assemblies. This benchmark comes with a splitting strategy into 80% for train, 10% for validation and 10% for test. The validation set is used for best method selection throughout the training epochs and the final performances presented in this section are obtained on the test set. Following previous works, we compare our approach on 3 seeds and report below the average performance. In addition, we construct a new OMC25-Cluster17 dataset from the OMC25 database (Gharakhanyan et al., 2025) and evaluate our models on this new dataset. We follow the same procedure as for COD-Cluster17 and describe the construction, in particular the filtering procedure of highly redundant data points, in Appendix E.1 which results in 43k structures.

**Model**  We primarily compare our method called *SinkFast* to the state-of-the-art AssembleFlow (Guo et al., 2025) method and thus reuse the same model. In particular, we consider here the atom-level SE(3)-equivariant model version described in Appendix C.2, that we refer to as the backbone. Then we will refer to AssembleFlow as this backbone trained with a flow matching scheme, while SinkFast refers to the same backbone trained with permutation invariance.

**Training methods**  AssembleFlow uses a flow matching setting, in which the model is trained on various interpolated rigid-body positions, which helps guide the optimization process. In contrast, our method *SinkFast* is trained in *simple regression*, which is defined as the task of predicting the target rigid-body positions directly from the initial positions. In this setting, the model takes exclusively as input the initial rigid-body positions. Simple regression is equivalent to a one-step flow matching.

While AssembleFlow is trained with $\mathcal{L}_{\mathrm{ML}}$ (Eq. 7) as the training objective, our method *SinkFast* is trained with one of the following objectives: $\mathcal{L}_{\mathrm{ML}}^*$ (Eq. 7), $\mathcal{L}_{\mathrm{RMSD}}^*$ (Eq. 9) or $\mathcal{L}_{\mathrm{Geom}}^*$ (Eq. 11) with permutation-invariance (Eq. 12). Standard hyperparameters and models' parameters are provided in Appendix C.2. In particular, we keep the hyperparameter $\alpha$ in Eq. 7 fixed to 10, as it was tuned by AssembleFlow authors for the task, and use 50 time steps of flow matching.

**Baselines**  Inorganic crystal structure prediction is a fast-moving domain in which many state-of-the-art models compete and innovate. We compare the performance of current organic state of the art to the

Table 1: Our best model *SinkFast* against state-of-the-art models on COD-Cluster17. Our method is trained with $\mathcal{L}^*_{\mathrm{ML}}$ (Eq. 7) or $\mathcal{L}^*_{\mathrm{RMSD}}$ (Eq. 9) with permutation-invariance (Eq. 12). The best results are marked in **bold**.

| | Pretrained Weights | Flow Matching | Packing Matching in Å↓ | | | |
|---|---|---|---|---|---|---|
| | | | $\mathrm{PM_{center}}$ | $\mathrm{PM_{atom}}$ | $\mathrm{PM^*_{center}}$ | $\mathrm{PM^*_{atom}}$ |
| Dataset: **COD-Cluster17-5K** | | | | | | |
| PackMol (Martínez et al., 2009) | | | $6.05_{\pm 0.05}$ | $7.10_{\pm 0.05}$ | - | - |
| CDVAE (Xie et al., 2022)[†] | | | - | - | $10.50_{\pm 0.52}$ | - |
| DiffCSP (Jiao et al., 2023)[†] | | | - | - | $23.50_{\pm 2.44}$ | - |
| Cond-CDVAE (Luo et al., 2024)[†] | ✓ | | - | - | $\mathbf{2.19_{\pm 0.03}}$ | - |
| DiffCSP (Jiao et al., 2023)[†] | ✓ | | - | - | $7.30_{\pm 3.65}$ | - |
| DiffCSP-MultMol (Jiao et al., 2023)[†] | ✓ | | - | - | $4.91_{\pm 1.19}$ | - |
| GNN-MD (Liu et al., 2024a) | | | $13.80_{\pm 0.07}$ | $13.67_{\pm 0.06}$ | - | - |
| CrystalFlow-LERP (Liu et al., 2024b) | | ✓ | $13.26_{\pm 0.09}$ | $13.59_{\pm 0.09}$ | - | - |
| AssembleFlow (Guo et al., 2025) | | ✓ | $6.13_{\pm 0.10}$ | $7.27_{\pm 0.04}$ | $3.86_{\pm 0.13}$ | $5.79_{\pm 0.012}$ |
| SinkFast - $\mathcal{L}^*_{\mathrm{ML}}$ *(ours)* | | | $\mathbf{5.80_{\pm 0.03}}$ | $\mathbf{6.96_{\pm 0.03}}$ | $\mathbf{3.60_{\pm 0.04}}$ | $\mathbf{5.54_{\pm 0.04}}$ |
| SinkFast - $\mathcal{L}^*_{\mathrm{RMSD}}$ *(ours)* | | | $\mathbf{5.85_{\pm 0.05}}$ | $\mathbf{6.98_{\pm 0.05}}$ | $3.77_{\pm 0.12}$ | $5.67_{\pm 0.08}$ |
| Dataset: **COD-Cluster17-All** | | | | | | |
| PackMol (Martínez et al., 2009) | | | $6.09_{\pm 0.01}$ | $7.15_{\pm 0.01}$ | - | - |
| GNN-MD (Liu et al., 2024a) | | | $14.51_{\pm 0.82}$ | $22.30_{\pm 12.04}$ | - | - |
| CrystalFlow-LERP (Liu et al., 2024b) | | ✓ | $13.28_{\pm 0.01}$ | $13.61_{\pm 0.00}$ | - | - |
| AssembleFlow (Guo et al., 2025) | | ✓ | $6.21_{\pm 0.01}$ | $7.37_{\pm 0.01}$ | $3.51_{\pm 0.05}$ | $5.60_{\pm 0.03}$ |
| SinkFast - $\mathcal{L}^*_{\mathrm{ML}}$ *(ours)* | | | $\mathbf{5.80_{\pm 0.00}}$ | $\mathbf{7.00_{\pm 0.01}}$ | $3.47_{\pm 0.04}$ | $\mathbf{5.51_{\pm 0.02}}$ |
| SinkFast - $\mathcal{L}^*_{\mathrm{RMSD}}$ *(ours)* | | | $\mathbf{5.80_{\pm 0.00}}$ | $\mathbf{7.00_{\pm 0.01}}$ | $\mathbf{3.41_{\pm 0.04}}$ | $5.54_{\pm 0.01}$ |

[†] Because these methods generate both the molecular conformation and molecular positions, they produce crystals of chemically different molecules wrt the ground truth. As a result, the $\mathrm{PM_{atom}}$ metric is not appropriate for assessing such approaches, since organic crystal structure prediction aims to reproduce the crystal structure of a specific molecular conformation. They are also excluded from evaluation on the COD-Cluster17-All dataset due to their prohibitively high computational cost.

inorganic one. As the latter models are not directly applicable to COD-Cluster17's problem, we conduct different experiments, either retraining both CDVAE (Xie et al., 2022) and DiffCSP (Jiao et al., 2023) models, or using pretrained weights. However, as CDVAE evaluates COD-Cluster17-5k test set in 25 hours, we do not evaluate it on COD-Cluster17-All test set being 22 times bigger. Implementation details are provided in Appendix E.2. Three other baselines, PackMol, GNN-MD and CrystalFlow-LERP, are motivated by the AssembleFlow paper (Guo et al., 2025), from which their results are extracted.

All models were trained on a single NVidia H100 GPU system, with 80GB memory and 67 TFlops. We trained them for 500 epochs, with a batch size of 8, Adam optimizer with a learning rate $10^{-4}$ adapted by a Cosine Annealing scheduler.

## 5.2 Main results

In Table 1 we present our models performance against six other state-of-the-art models on the COD-Cluster17 dataset. First, while original state-of-the-art results are presented in $\mathrm{PM_{center}}$ and $\mathrm{PM_{atom}}$, we show under $\mathrm{PM^*_{center}}$ and $\mathrm{PM^*_{atom}}$ the importance of optimal assignment to get the best metric performance over the set of $\mathcal{S}$-permutations of the predicted assembly as detailed in section 4.3. Indeed the metric decreases greatly under this optimal assignment, indicating the inability of models to memorize positions for each molecule as they are equivalent. Then, we show that our method outperforms all other baselines on both 5k subset and the full dataset by a significant margin. Notably our method is the only molecule-level deep learning method that outperforms PackMol on both datasets. The results of atom-level models CDVAE (Xie et al., 2022), Cond-CDVAE (Luo et al., 2024) and DiffCSP (Jiao et al., 2023) on the COD-Cluster17-5k test set show that enforcing symmetry and periodic boundary constraints through lattice prediction is a very promising

Table 2: Ablation study of using physically grounded losses during training on COD-Cluster17 in two different training schemes: flow matching or simple regression with permutation-invariant loss.

| Loss | Flow Matching | Test Loss in Å↓ | | Packing matching in Å↓ | | | |
|---|---|---|---|---|---|---|---|
| | | $\mathcal{L}^*_{\mathrm{RMSD}}$ | $\mathcal{L}^*_{\mathrm{Geom}}$ | $\mathrm{PM}^*_{\mathrm{center}}$ | $\mathrm{PM}^*_{\mathrm{atom}}$ | $\mathrm{PM}_{\mathrm{center}}$ | $\mathrm{PM}_{\mathrm{atom}}$ |
| Dataset: **COD-Cluster17-5K** | | | | | | | |
| $\mathcal{L}_{\mathrm{ML}}$ | ✓ | $9.56_{\pm 0.07}$ | $13.35_{\pm 0.30}$ | $3.86_{\pm 0.13}$ | $5.79_{\pm 0.12}$ | $6.28_{\pm 0.15}$ | $7.39_{\pm 0.16}$ |
| $\mathcal{L}_{\mathrm{RMSD}}$ | ✓ | $9.41_{\pm 0.19}$ | $13.37_{\pm 0.24}$ | $3.89_{\pm 0.16}$ | $5.80_{\pm 0.16}$ | $6.26_{\pm 0.20}$ | $7.35_{\pm 0.21}$ |
| $\mathcal{L}_{\mathrm{Geom}}$ | ✓ | $9.22_{\pm 0.06}$ | $10.45_{\pm 0.55}$ | $3.90_{\pm 0.08}$ | $5.85_{\pm 0.04}$ | $6.14_{\pm 0.02}$ | $7.20_{\pm 0.05}$ |
| $\mathcal{L}^*_{\mathrm{ML}}$ | | $\mathbf{8.69_{\pm 0.06}}$ | $12.16_{\pm 0.12}$ | $\mathbf{3.60_{\pm 0.04}}$ | $\mathbf{5.54_{\pm 0.04}}$ | $\mathbf{5.80_{\pm 0.03}}$ | $\mathbf{6.96_{\pm 0.03}}$ |
| $\mathcal{L}^*_{\mathrm{RMSD}}$ | | $\mathbf{8.73_{\pm 0.07}}$ | $12.05_{\pm 0.15}$ | $3.77_{\pm 0.12}$ | $5.67_{\pm 0.08}$ | $\mathbf{5.85_{\pm 0.05}}$ | $\mathbf{6.98_{\pm 0.05}}$ |
| $\mathcal{L}^*_{\mathrm{Geom}}$ | | $9.32_{\pm 0.06}$ | $\mathbf{8.78_{\pm 0.05}}$ | $5.55_{\pm 0.15}$ | $6.54_{\pm 0.07}$ | $6.92_{\pm 0.07}$ | $7.46_{\pm 0.02}$ |
| Dataset: **COD-Cluster17-All** | | | | | | | |
| $\mathcal{L}_{\mathrm{ML}}$ | ✓ | $9.26_{\pm 0.18}$ | $12.02_{\pm 0.30}$ | $3.51_{\pm 0.05}$ | $5.60_{\pm 0.03}$ | $5.96_{\pm 0.02}$ | $7.15_{\pm 0.03}$ |
| $\mathcal{L}_{\mathrm{RMSD}}$ | ✓ | $9.08_{\pm 0.12}$ | $12.17_{\pm 0.33}$ | $3.51_{\pm 0.04}$ | $5.60_{\pm 0.03}$ | $5.97_{\pm 0.05}$ | $7.18_{\pm 0.05}$ |
| $\mathcal{L}_{\mathrm{Geom}}$ | ✓ | $9.33_{\pm 0.10}$ | $10.77_{\pm 0.13}$ | $3.78_{\pm 0.09}$ | $5.76_{\pm 0.05}$ | $6.09_{\pm 0.07}$ | $7.21_{\pm 0.05}$ |
| $\mathcal{L}^*_{\mathrm{ML}}$ | | $\mathbf{8.65_{\pm 0.02}}$ | $12.10_{\pm 0.10}$ | $\mathbf{3.47_{\pm 0.04}}$ | $\mathbf{5.51_{\pm 0.02}}$ | $\mathbf{5.80_{\pm 0.00}}$ | $\mathbf{7.00_{\pm 0.01}}$ |
| $\mathcal{L}^*_{\mathrm{RMSD}}$ | | $\mathbf{8.70_{\pm 0.03}}$ | $12.16_{\pm 0.08}$ | $\mathbf{3.41_{\pm 0.04}}$ | $\mathbf{5.54_{\pm 0.01}}$ | $\mathbf{5.80_{\pm 0.00}}$ | $\mathbf{7.00_{\pm 0.01}}$ |
| $\mathcal{L}^*_{\mathrm{Geom}}$ | | $9.35_{\pm 0.00}$ | $\mathbf{8.71_{\pm 0.03}}$ | $5.43_{\pm 0.10}$ | $6.52_{\pm 0.05}$ | $6.84_{\pm 0.06}$ | $7.45_{\pm 0.02}$ |
| Dataset: **OMC25-Cluster17** | | | | | | | |
| *Baseline* | | $16.34_{\pm 0.03}$ | $10.53_{\pm 0.01}$ | $5.19_{\pm 0.08}$ | $6.50_{\pm 0.10}$ | $6.78_{\pm 0.09}$ | $7.45_{\pm 0.11}$ |
| $\mathcal{L}_{\mathrm{ML}}$ | ✓ | $8.47_{\pm 0.08}$ | $11.49_{\pm 0.39}$ | $\mathbf{3.56_{\pm 0.06}}$ | $\mathbf{4.92_{\pm 0.05}}$ | $\mathbf{5.37_{\pm 0.10}}$ | $\mathbf{6.24_{\pm 0.08}}$ |
| $\mathcal{L}_{\mathrm{RMSD}}$ | ✓ | $8.45_{\pm 0.04}$ | $11.41_{\pm 0.28}$ | $\mathbf{3.55_{\pm 0.04}}$ | $5.00_{\pm 0.03}$ | $\mathbf{5.36_{\pm 0.05}}$ | $\mathbf{6.27_{\pm 0.05}}$ |
| $\mathcal{L}_{\mathrm{Geom}}$ | ✓ | $14.38_{\pm 0.04}$ | $10.83_{\pm 0.62}$ | $4.20_{\pm 0.30}$ | $5.35_{\pm 0.16}$ | $5.82_{\pm 0.11}$ | $6.53_{\pm 0.06}$ |
| $\mathcal{L}^*_{\mathrm{ML}}$ | | $\mathbf{8.17_{\pm 0.00}}$ | $10.72_{\pm 0.02}$ | $3.59_{\pm 0.03}$ | $5.00_{\pm 0.02}$ | $\mathbf{5.34_{\pm 0.00}}$ | $\mathbf{6.21_{\pm 0.01}}$ |
| $\mathcal{L}^*_{\mathrm{RMSD}}$ | | $\mathbf{8.12_{\pm 0.01}}$ | $10.95_{\pm 0.11}$ | $3.57_{\pm 0.09}$ | $5.01_{\pm 0.05}$ | $\mathbf{5.32_{\pm 0.02}}$ | $\mathbf{6.20_{\pm 0.01}}$ |
| $\mathcal{L}^*_{\mathrm{Geom}}$ | | $14.47_{\pm 0.73}$ | $\mathbf{8.90_{\pm 0.05}}$ | $7.76_{\pm 0.16}$ | $7.82_{\pm 0.12}$ | $8.20_{\pm 0.14}$ | $8.12_{\pm 0.10}$ |

direction. However, these models predict atomic positions individually for a single ASU which leads to chemically different molecules from the target ones.

## 5.3 Ablation studies

**Training with introduced losses** Table 2 lists experiments conducted when training with physically grounded losses, both in flow matching as in AssembleFlow (Guo et al., 2025) and in our simple regression model with permutation invariant loss. We can draw 2 main conclusions: (1) training with $\mathcal{L}_{\mathrm{RMSD}}$ (Eq. 9) performs on par with training with the tuned standard $\mathcal{L}_{\mathrm{ML}}$ (Eq. 7) while having no parameter to tune. And (2) both previous absolute losses fail to perform on the geometric relative packing loss metric $\mathcal{L}^*_{\mathrm{Geom}}$ (Eq. 11) while training with it is the solution. In reverse, training with this relative loss yields poor results on absolute ones. This mainly shows the limitations of the absolute packing matching task of interest here.

Moreover, these results along with ablation in Appendix D show a significant gain, dividing by up to 3.8 the evaluation metric, when training with the $\mathcal{S}$-permutation invariant losses, while not being useful in flow matching. This reveals a redundancy between the two and that the main interest of flow matching on this task thus lies in being an optimal transport approximator. While the added value of flow matching is yet to be proven, its usage out of the box may not come handy. We believe it should be better adapted to the domain specificities and tasks in further studies.

As rigid body transformations are composed here of both a rotation and a translation individually predicted by our model, we present in Appendix D.5 the individual pure $\mathbb{R}^3$ and SO(3) performances. This experiment shows that both AssembleFlow and SinkFast focus on positioning molecules in space while mostly discarding their orientation.

Table 3: Execution times for our method reported on COD-Cluster17-5k on a single NVidia H100 GPU system, with 80GB memory and 67 TFlops. Results are presented as average over 10 epochs at training and over 10 batches at inference. AssembleFlow is trained with 50 timesteps.

| Method | Training time (s) per epoch | Test time (s) per batch |
|---|---|---|
| Cond-CDVAE - *pretrained* | - | 5760 |
| AssembleFlow | 2678.5 | 0.89 |
| SinkFast - *without permutation-invariant loss* | 53.6 | 0.02 |
| SinkFast - *with permutation-invariant loss* | 64.1 | 0.02 |

**Execution time**    Table 3 lists the execution time for the different methods. In particular, we are interested in the expense of our $\mathcal{S}$-permutation invariant loss and how it compares to the cost of using a flow matching scheme. While it increases the training execution time by 20% compared to simple regression without the permutation invariance, it saves by a factor 42 the overall training time compared to flow matching. And as it is only used during training, the gain at inference is a factor 50 (number of time steps) compared to flow matching. However, as only one timestep is usually sampled at training in flow matching, it is better to compare convergence time in terms of number of epochs before convergence. While AssembleFlow converges after 350 epochs, SinkFast converges after less than 100 epochs. The actual gain at training time is then a factor 3.5. As mentioned in Guo et al. (2025), PackMol is 25 times slower than AssembleFlow, which makes it about 1,000 times slower than SinkFast. We also evaluate the pretrained Cond-CDVAE (Luo et al., 2024) on the COD-Cluster17-5k test set, which executes in 96 minutes for a single batch.

## 6    Limitations

**Prediction quality**    Although our objective function refinement helps to boost the reported metrics, the visualizations reported in Appendix F also show the large performance gap remaining to be closed in discovering plausible and stable crystal structures. In particular, the orientations of molecules are highly incorrect as reported and discussed in the ablation study in Appendix D.5. We believe new methods should make use of all the geometrical properties of materials science to design powerful yet efficient algorithms that can reliably perform on tasks always closer to real-life data. To the best of our knowledge, this limitation affects all published work on the topic, including ours, highlighting the current boundaries of the field and future research challenges.

**Generalization**    While this work pushes the frontier of materials discovery on a specific benchmark, its usefulness to other benchmarks is yet to be assessed. Our work has been designed to tackle a weakness in the problem definition of common molecular assembly tasks and highlights the need for a revised dataset definition. With a real-life application in mind, the absence of periodic boundary conditions is a fundamental limitation of the COD-Cluster17 dataset and thus to this method. Indeed, a predicted molecule position should be correct up to any unit cell translation. However, as no periodicity information is available, prediction has to match absolute target positions, which hinders the generalisation capability of any model. A second major limitation in COD-Cluster17 is the absence of space group information for each training sample. The same rigid molecule can crystallize in different configurations according to specific symmetry groups inside a unit cell that then replicates infinitely in space. This conditions global structure prediction – and thus also the subtask of molecules assembly – and can give different targets for the same common data. As a result, the generalisation capability of any model is greatly hindered.

**Applicability**    On the one hand, the molecule assembly task of COD-Cluster17 is far from the non-polymeric crystal structure prediction one. While it is the only available benchmark in this field, the community needs a more physically meaningful dataset with a proper benchmark definition, based either on COD (Gražulis et al., 2009) (about 500k crystals) or OMC25 (Gharakhanyan et al., 2025) (about 25 million crystals). However, OMC25 consists of simulated zero-temperature structures, unlike the experimentally observed COD data.

On the other hand, if we want to use our model in practice, the molecular conformation is usually not known and deeply related to the crystal structure. We study in Appendix E.3 the rigid body approximation and our model's performance on a different real-world test set, in which we generate new molecular conformations through RDKit (Landrum et al., 2025). We conclude that the current approximation is valid, however future models should be trained end-to-end, jointly learning conformation and crystal structure prediction.

## 7 Conclusion

In this paper we have focused on a simpler subtask of the complex organic crystal structure prediction. We have shown the necessity to use meaningful metrics on a benchmark and proven its utmost importance to accurately compare methods. We have also demonstrated the importance to train models with rigid-body losses grounded in physical principles that greatly improve performance on molecule assembly. Such metrics are also essential to assess real-world applicability of current methods. Our main contribution is the demonstration that the appropriate definition of a meaningful learning objective simplifies the problem, boosts the performance and speeds up the training scheme. We release a simple implementation of the method to be used in future benchmarks. This work invites to take a step back from large generative models and expensive methods, and instead focus on proper problem definition and principled, physics-inspired solutions.

## Acknowledgements

EJ, RM and JM were supported by the ERC grant number 101087696 (APHELEIA project). SG has been partially supported by state aid managed by the National Research Agency under the France 2030 program (Grant ANR-23-IACL-0006). This work was granted access to the HPC resources of IDRIS under the allocation 2025-AD011014006R2 made by GENCI. We thank Kliment Olechnovič for his help with molecular visualization, Roman Klypa for his support with equivariant models and Romain Séailles for his help with the optimization techniques. We are particularly grateful to Saulius Grazulis from Vilnius University for guiding us through the COD database and sharing with us his valuable expertise.

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

## Appendix

*We release a version of the code available at [https://github.com/inria-thoth/SinkFast](https://github.com/inria-thoth/SinkFast).*

## A    RMSD and Rigid Motions

### A.1    Notations

Throughout this section we will be generally dealing with $3 \times 3$ matrices and 3-vectors. Therefore, for linear algebra operations we will stick to the following notation. Bold upper case letters (i.e., $\mathbf{A}$) will denote matrices, normal weight lower case letters (i.e., $c$) will denote scalars, and we will also use an arrow notation for 3-vectors, such as $\vec{v}$. Most of the information reported here can be found in the original papers that deal with rigid-body measures for rigid molecules by Popov & Grudinin (2014); Pagès et al. (2018); Pagès & Grudinin (2018).

### A.2    Rigid-body arithmetic

As we introduced in the main text, a rigid spatial transformation operator $\mathcal{T} = (\vec{t}, Q)$ is composed of a 3D translation $\vec{t} \in \mathbb{R}^3$ and a 3D rigid rotation quaternion $Q = [s, \vec{q}] \in \mathrm{SO}(3)$, which can also be represented with a rotation matrix $\mathbf{R}$, such that $\mathcal{T} = (\vec{t}, \mathbf{R})$. It is useful to introduce a composition of spatial transformation operators $\mathcal{T}_2 \circ \mathcal{T}_1$, where the operator $\mathcal{T}_1$ on the right is applied first, and an inverse $\mathcal{T}^{-1}$. The composition will be given as

$$\mathcal{T}_2 \circ \mathcal{T}_1 = (\vec{t}_2 + \mathbf{R}_2 \vec{t}_1, \mathbf{R}_2 \mathbf{R}_1) \equiv (\vec{t}_2 + Q_2 \cdot \vec{t}_1, Q_2 \cdot Q_1), \tag{A.1}$$

where we define the qurternion product in the next section. The inverse will be:

$$\mathcal{T}^{-1} = (-\mathbf{R}^{-1}\vec{t}, \mathbf{R}^{-1}) \equiv (-\mathbf{R}^T \vec{t}, \mathbf{R}^T) \equiv (-Q^{-1} \cdot \vec{t}, Q^{-1}), \tag{A.2}$$

with an inverse qurternion defined below.

### A.3    Quaternion arithmetic

It is very convenient to express three-dimensional rotations using quaternion arithmetic. Thus, we will give a brief summary of it here. We consider a quaternion $Q$ as a combination of a scalar $s$ with a 3-component vector $\vec{q} = \{q_x, q_y, q_z\}$, $Q = [s, \vec{q}]$. Quaternion algebra defines multiplication, division, inversion and norm, among other operations. The product of two quaternions $Q_1 = [s_1, \vec{q}_1]$ and $Q_2 = [s_2, \vec{q}_2]$ is a quaternion and can be expressed through a combination of scalar and vector products:

$$Q_1 \cdot Q_2 \equiv [s_1, \vec{q}_1] \cdot [s_2, \vec{q}_2] = [s_1 s_2 - (\vec{q}_1 \cdot \vec{q}_2), s_1 \vec{q}_2 + s_2 \vec{q}_1 + (\vec{q}_1 \times \vec{q}_2)]. \tag{A.3}$$

The squared norm of a quaternion $Q$ is given as $|Q|^2 = s^2 + \vec{q} \cdot \vec{q}$, and a unit quaternion $\hat{Q}$ is a quaternion with its norm equal to 1. An inverse quaternion $Q^{-1}$ is given as $Q^{-1} = [s, -\vec{q}]/|Q|^2$. A vector $\vec{v}$ can be treated as a quaternion with a zero scalar component, $\vec{v} \equiv [0, \vec{v}]$. Then, a unit quaternion $\hat{Q}$ can be used to rotate vector $\vec{v}$ to a new position $\vec{v}'$ as follows

$$[0, \vec{v}'] = \hat{Q}[0, \vec{v}]\hat{Q}^{-1} = \left[0, (s^2 - \vec{q}^2)\vec{v} + 2s(\vec{q} \times \vec{v}) + 2(\vec{q} \cdot \vec{v})\vec{q}\right] = [0, \vec{v} + 2\vec{q} \times (\vec{q} \times \vec{v} + s\vec{v})]. \tag{A.4}$$

Equivalently, the same rotation can be represented with a rotation matrix $\mathbf{R}$, such that $\vec{v}' = \mathbf{R}\vec{v}$, where $\mathbf{R}$ can be expressed through the components of the quaternion $\hat{Q}$ as

$$\mathbf{R} = \begin{pmatrix} s^2 + q_x^2 - q_y^2 - q_z^2 & 2q_x q_y - 2sq_z & 2q_x q_z + 2sq_y \\ 2q_x q_y + 2sq_z & s^2 - q_x^2 + q_y^2 - q_z^2 & 2q_y q_z - 2sq_x \\ 2q_x q_z - 2sq_y & 2q_y q_z + 2sq_x & s^2 - q_x^2 - q_y^2 + q_z^2 \end{pmatrix}. \tag{A.5}$$

A unit quaternion $\hat{Q}$ corresponding to a rotation by an angle $\alpha$ around a unit axis $\vec{u}$ is given as $\hat{Q} = [\cos\frac{\alpha}{2}, \vec{u}\sin\frac{\alpha}{2}]$, and its inverse is $\hat{Q}^{-1} = [\cos\frac{\alpha}{2}, -\vec{u}\sin\frac{\alpha}{2}]$. Finally, $N$ sequential rotations around different unit axes defined by unit quaternions $\{\hat{Q}_i\}_N$ result in a new vector $\vec{v}'$ according to

$$[0, \vec{v}'] = \hat{Q}_N \hat{Q}_{N-1} ... \hat{Q}_2 \hat{Q}_1 \, [0, \vec{v}] \, \hat{Q}_1^{-1} \hat{Q}_2^{-1} ... \hat{Q}_{N-1}^{-1} \hat{Q}_N^{-1}. \tag{A.6}$$

### A.4  Root mean square deviation

The root mean square deviation (RMSD) is one of the most widely used similarity criteria in chemistry, structural biology, bioinformatics, and material science. We will stick to this measure here, as it is very powerful, easy to understand and also because it can be computed very efficiently. For our particular needs we will use the definition of RMSD between two ordered sets of points, where each point has an equal contribution to the overall RMSD loss. More precisely, given a set of $N$ points $A = \{\vec{a}_i\}_N$ and $B = \{\vec{b}_i\}_N$ with associated weights $w = \{w_i\}_N$, the RMSD between them is defined as

$$\mathrm{RMSD}(A, B)^2 = \frac{1}{W} \sum_{1 \leq i \leq N} w_i \left| \vec{a}_i - \vec{b}_i \right|^2, \tag{A.7}$$

where $W = \sum_i w_i$. Here, $\{w_i\}_N$ are statistical weights that may emphasize the importance of a certain part of the molecular structure, for example in case of a protein, the backbone or $C_\alpha$ atoms. These weights can also be equal to atomic masses (in this case $W$ equals to the total mass of the molecule) or may be set to unity (in this case $W = N$). In this work, we set the weights to unity, thus

$$\mathrm{RMSD}(A, B)^2 = \frac{1}{N} \sum_{1 \leq i \leq N} \left| \vec{a}_i - \vec{b}_i \right|^2, \tag{A.8}$$

since it makes the following equations simpler to read and to use in practice. However, we should keep in mind that the weights can be easily added to all the corresponding equations.

### A.5  Rigid body motion described with quaternions

Let $\mathbf{R}$ be a rotation matrix and $\vec{t}$ a translation vector applied to a molecule with $N$ atoms at positions $A = \{\vec{a}_i\}_N$ with $\vec{a}_i = \{x_i, y_i, z_i\}^T$, such that the new positions $A' = \{\vec{a}_i'\}_N$ are given as $\vec{a}_i' = \mathbf{R}\vec{a}_i + \vec{t}$. Then, the weighted RMSD between $A$ and $A'$ will be given as

$$\mathrm{RMSD}^2(A, A') = \frac{1}{W} \sum_i w_i \left| \vec{a}_i - \mathbf{R}\vec{a}_i - \vec{t} \right|^2. \tag{A.9}$$

We can rewrite the previous expression using quaternion representation of vectors $\vec{a}_i$ and $\vec{t}$ as

$$\mathrm{RMSD}^2 = \frac{1}{W} \sum_i w_i \left| [0, \vec{a}_i] - \hat{Q}[0, \vec{a}_i]\hat{Q}^{-1} - [0, \vec{t}] \right|^2. \tag{A.10}$$

Here, the unit quaternion $\hat{Q}$ corresponds to the rotation matrix $\mathbf{R}$. Since the norm of a quaternion does not change if we multiply it by a unit quaternion, we may right-multiply the kernel of the previous expression by $\hat{Q}$ to obtain

$$\mathrm{RMSD}^2 = \frac{1}{W} \sum_i w_i \left| [0, \vec{a}_i]\hat{Q} - \hat{Q}[0, a_i] - [0, \vec{t}]\hat{Q} \right|^2. \tag{A.11}$$

Using the scalar–vector representation of a quaternion, $\hat{Q} = [s, \vec{q}]$, we rewrite the previous RMSD expression as

$$\mathrm{RMSD}^2 = \frac{1}{W} \sum_i w_i \left[ -\vec{q} \cdot \vec{t}, -s\vec{t} + (2\vec{a}_i - \vec{t}) \times \vec{q} \right]^2. \tag{A.12}$$

Performing scalar and vector products in Eq. (A.12), we obtain

$$
\begin{aligned}
\text{RMSD}^2 \quad = \quad & \frac{1}{W} \sum_i w_i \Big( [q_x t_x + q_y t_y + q_z t_z]^2 \\
& + \quad [-s t_x + q_y(2 z_i - t_z) - q_z(2 y_i - t_y)]^2 \\
& + \quad [-s t_y + q_z(2 x_i - t_x) - q_x(2 z_i - t_z)]^2 \\
& + \quad [-s t_z + q_x(2 y_i - t_y) - q_y(2 x_i - t_x)]^2 \Big).
\end{aligned}
\tag{A.13}
$$

Grouping terms in Eq. (A.13) that depend on atomic positions together, we obtain

$$
\begin{aligned}
\text{RMSD}^2 \quad = \quad & t_x^2 + t_y^2 + t_z^2 + \frac{4}{W} \sum_i w_i \{ q_x^2(y_i^2 + z_i^2) + q_y^2(x_i^2 + z_i^2) + q_z^2(x_i^2 + y_i^2) \\
& - \quad 2 q_x q_y x_i y_i - 2 q_x q_z x_i z_i - 2 q_y q_z z_i y_i \} \\
& + \quad \frac{4}{W} \left\{ q_x q_z t_z + q_x q_y t_y - q_z^2 t_x - q_y^2 t_x + s q_z t_y - s q_y t_z \right\} \sum_i w_i x_i \\
& + \quad \frac{4}{W} \left\{ q_y q_z t_z + q_x q_y t_x - q_x^2 t_y - q_z^2 t_y + s q_x t_z - s q_z t_x \right\} \sum_i w_i y_i \\
& + \quad \frac{4}{W} \left\{ q_y q_z t_y + q_x q_z t_x - q_x^2 t_z - q_y^2 t_z + s q_y t_x - s q_x t_y \right\} \sum_i w_i z_i.
\end{aligned}
\tag{A.14}
$$

Introducing the inertia tensor $\mathbf{I}$, the rotation matrix $\mathbf{R}$, the center of mass vector $\vec{c}$, and the $3 \times 3$ identity matrix $\mathbf{E}_3$, we may simplify the previous expression to

$$
\text{RMSD}^2 = \vec{t}^2 + \frac{4}{W} \vec{q}^T \mathbf{I} \vec{q} + 2 \vec{t}^T (\mathbf{R} - \mathbf{E}_3) \vec{c},
\tag{A.15}
$$

where $\vec{c} = \frac{1}{W} \{ \sum w_i x_i, \sum w_i y_i, \sum w_i z_i \}^T$, rotation matrix $\mathbf{R}$ corresponds to the rotation with the unit quaternion $\hat{Q}$ according to Eq. (A.5), and the inertia tensor $\mathbf{I}$ is given as

$$
\mathbf{I} = \begin{pmatrix}
\sum w_i(y_i^2 + z_i^2) & -\sum w_i x_i y_i & -\sum w_i x_i z_i \\
-\sum w_i x_i y_i & \sum w_i(x_i^2 + z_i^2) & -\sum w_i y_i z_i \\
-\sum w_i x_i z_i & -\sum w_i y_i z_i & \sum w_i(x_i^2 + y_i^2)
\end{pmatrix}.
\tag{A.16}
$$

The RMSD expression (A.15) consists of three parts, the pure translational contribution $\vec{t}^2$, the pure rotational contribution $\frac{4}{W} \vec{q}^T \mathbf{I} \vec{q}$, and the cross term $2 \vec{t}^T (\mathbf{R} - \mathbf{E}_3) \vec{c}$. In this equation, only two variables depend on the atomic positions $\{\vec{a}_i\}_N$, the inertia tensor $\mathbf{I}$, and the center of mass vector $\vec{c}$. These depend only on the reference structure of a rigid molecule, and can be precomputed. Moreover, it is practical to choose a reference frame centred on the molecular center of mass. In this frame, the cross term vanishes and the above RMSD equation simplifies to

$$
\text{RMSD}^2 = \vec{t}^2 + \frac{4}{W} \vec{q}^T \mathbf{I} \vec{q}.
\tag{A.17}
$$

However, we must bring reader's attention that the inertia tensor must be specifically computed in the chosen reference frame.

### A.6 SE(3) flow matching

**In the Euclidean space** Conditional Flow Matching (Liu et al., 2023; Lipman et al., 2023; Albergo & Vanden-Eijnden, 2023) is a simple scalable method to train generative models. The basic principle is to choose a family $\mathbf{X} = \{(X_t)_{t \in [0,1]}\}$ of interpolating paths between any source distribution $\mathbb{P}_0$ and the target distribution $\mathbb{P}_1$. The paths should be differentiable and have their marginal laws at both ends $t = 0$ and $t = 1$ match the source and target distributions: $\mathcal{L}(X_0) = \mathbb{P}_0$ and $\mathcal{L}(X_1) = \mathbb{P}_1$, respectively. The flow matching procedure consists in training a neural network $u$ to match the conditional velocity field $v^{\mathbf{X}}$ induced by these paths:

$$v^{\mathbf{X}}(t, x) = \mathbb{E}\left[\dot{X}_t | X_t = x\right]. \tag{A.18}$$

In practice, this family path is created with linear interpolations (LERP) between samples $X_0, X_1$ from $\mathbb{P}_0, \mathbb{P}_1$:

$$X_t = (1 - t)X_0 + tX_1 = \text{LERP}(X_0, X_1, t). \tag{A.19}$$

At inference time, samples are generated by solving the forward ODE induced by the velocity field, by Euler discretization for example.

**In SO(3)**   While this framework was originally designed for $\mathbb{R}^d$, there exists an extension to SO(3) (Chen & Lipman, 2024). Indeed, by representing rotations with unit quaternions, there is a natural equivalent to linear interpolation, called Spherical Linear Interpolation (SLERP) (Shoemake, 1985). This creates differentiable interpolation paths $(\mathbf{q}_t)$ between source and target quaternions $\mathbf{q}_0, \mathbf{q}_1$:

$$\mathbf{q}_t = \text{SLERP}(\mathbf{q}_0, \mathbf{q}_1; t) = \mathbf{q}_0(\mathbf{q}_0^1 \mathbf{q}_1)^t. \tag{A.20}$$

Combining LERP and SLERP, it is possible to linearly interpolate between two rigid-body transformations $\mathcal{T}_0 = (\vec{r}_0, \mathbf{q}_0)$ and $\mathcal{T}_1 = (\vec{r}_1, \mathbf{q}_1)$ as $\mathcal{T}_t = (\text{LERP}(\vec{r}_0, \vec{r}_1, t), \text{SLERP}(\mathbf{q}_0, \mathbf{q}_1; t))$.

### A.7   Illustration of the proposed physically-grounded losses

We illustrate in Figure A.1 how the different proposed losses evolve when the prediction is similar to the ground-truth up to a certain rigid-body transformation, either rotation, translation or permutation. In each of these cases, the predicted structure is correct chemically and physically and the loss should thus be 0. This figure helps us illustrate 3 main motivations. First, the difference between the parameter dependent $\mathcal{L}_{\text{ML}}$ and the physically meaningful $\mathcal{L}_{\text{RMSD}}$. Second, the geometric loss is invariant to SE(3) transformations of the global picture but is not invariant to the index permutation of the arbitrarily chosen ordering of identical molecules. Third, this invariance to index permutation is enabled through the use of the linear sum assignment problem as detailed in section 4.1.

## B   Metrics

**Theorem B.1.** $PM_{atom}^2 \leq 2RMSD_{atom}^2$

*Proof.* Let us first define two metrics $\text{PM}_{\text{atom}}$ and $\text{RMSD}_{\text{atom}}$ as

$$\text{PM}_{\text{atom}}^2 = \frac{1}{n_{\text{atom}}^2} \sum_{i \in n_{\text{atom}}} \sum_{j \in n_{\text{atom}}} (||\vec{x}_{i,\text{pred}} - \vec{x}_{j,\text{pred}}|| - ||\vec{x}_{i,\text{gt}} - \vec{x}_{j,\text{gt}}||)^2, \tag{B.1}$$

$$\text{RMSD}_{\text{atom}}^2 = \frac{1}{n_{\text{atom}}} \sum_{i \in n_{\text{atom}}} ||\vec{x}_{i,\text{pred}} - \vec{x}_{i,\text{gt}}||^2. \tag{B.2}$$

We also define $\bar{x}_{\text{pred}} = \frac{1}{n_{\text{atom}}} \sum_i \vec{x}_{i,\text{pred}}$, $\bar{x}_{\text{gt}} = \frac{1}{n_{\text{atom}}} \sum_i \vec{x}_{i,\text{gt}}$, and use $\cdot$ as the scalar product. Let us write down the following expression,

$$\text{PM}_{\text{atom}}^2 - 2\text{RMSD}_{\text{atom}}^2 = \frac{1}{n_{\text{atom}}^2} \sum_{i \in n_{\text{atom}}} \sum_{j \in n_{\text{atom}}} \left( 4\vec{x}_{i,\text{pred}} \cdot \vec{x}_{i,\text{gt}} - 2(\vec{x}_{i,\text{pred}} \cdot \vec{x}_{j,\text{pred}} + \vec{x}_{i,\text{gt}} \cdot \vec{x}_{j,\text{gt}} \right.$$
$$\left. + ||\vec{x}_{i,\text{pred}} - \vec{x}_{j,\text{pred}}|| ||\vec{x}_{i,\text{gt}} - \vec{x}_{j,\text{gt}}||) \right)$$
$$= 4\overline{(x_{\text{pred}} \cdot x_{\text{gt}})} - 2\bar{x}_{\text{pred}} \cdot \bar{x}_{\text{pred}} - 2\bar{x}_{\text{gt}} \cdot \bar{x}_{\text{gt}} - \frac{2}{n_{\text{atom}}^2} \sum_{i \in n_{\text{atom}}} \sum_{j \in n_{\text{atom}}} ||\vec{x}_{i,\text{pred}} - \vec{x}_{j,\text{pred}}|| ||\vec{x}_{i,\text{gt}} - \vec{x}_{j,\text{gt}}|| \tag{B.3}$$

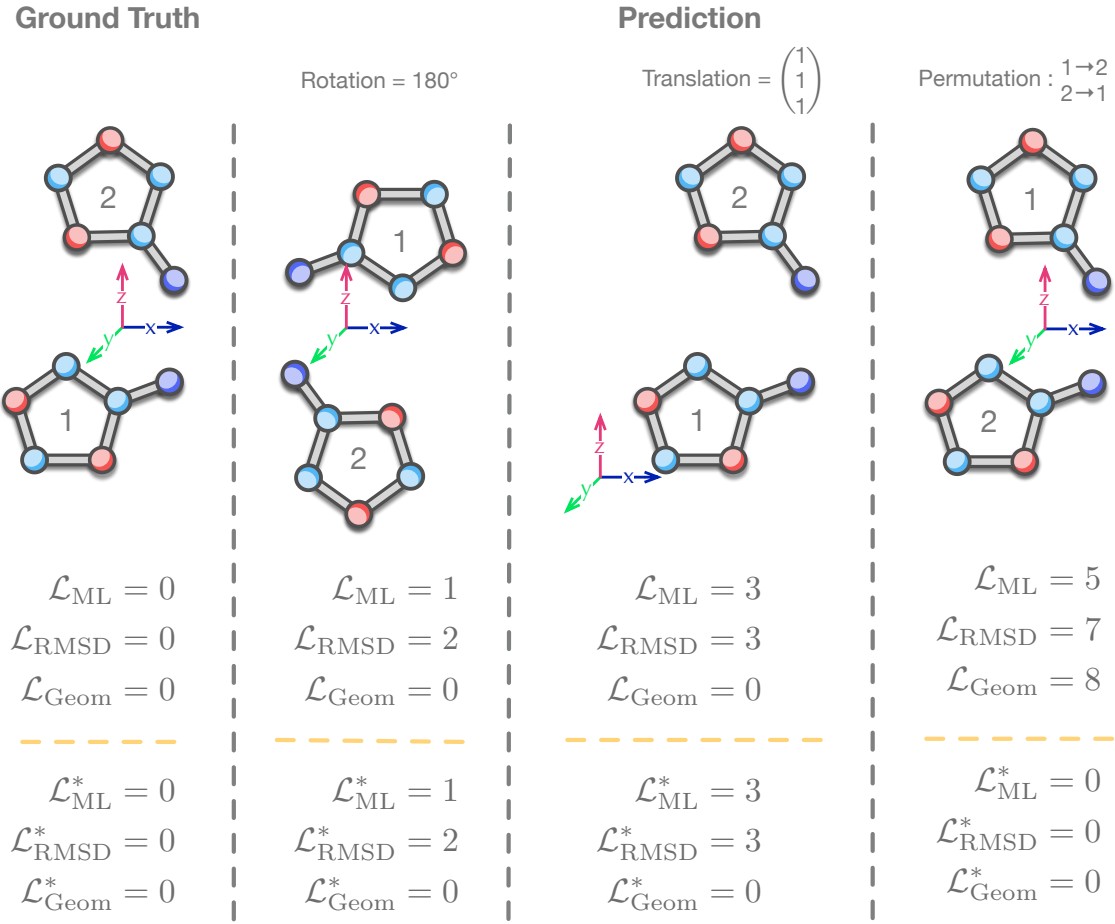

Figure A.1: Illustration how the proposed physically-grounded losses evolve under some transformations on a toy example. The numbers are arbitrary and not physically related.

By Cauchy-Schwarz enquality (or maximizing the cosine of an angle between two vectors), we obtain

$$
\frac{2}{n_{\text{atom}}^2} \sum_{i \in n_{\text{atom}}} \sum_{j \in n_{\text{atom}}} ||\vec{x}_{i,\text{pred}} - \vec{x}_{j,\text{pred}}|| ||\vec{x}_{i,\text{gt}} - \vec{x}_{j,\text{gt}}|| \geq
$$
$$
\frac{2}{n_{\text{atom}}^2} \sum_{i \in n_{\text{atom}}} \sum_{j \in n_{\text{atom}}} (\vec{x}_{i,\text{pred}} - \vec{x}_{j,\text{pred}}) \cdot (\vec{x}_{i,\text{gt}} - \vec{x}_{j,\text{gt}}),
\tag{B.4}
$$

which gives

$$
\text{PM}_{\text{atom}}^2 - 2\text{RMSD}_{\text{atom}}^2 \leq -2(\bar{\vec{x}}_{\text{pred}} - \bar{\vec{x}}_{\text{gt}})^2,
\tag{B.5}
$$

thus,

$$
\text{PM}_{\text{atom}}^2 - 2\text{RMSD}_{\text{atom}}^2 \leq 0.
\tag{B.6}
$$

$\square$

**Theorem B.2.** *PM metric is SE3-invariant.*

*Proof.* Let us again consider

$$\mathrm{PM}^2(x_{\mathrm{pred}}, x_{\mathrm{gt}}) = \frac{1}{n_{\mathrm{atom}}^2} \sum_{i \in n_{\mathrm{atom}}} \sum_{j \in n_{\mathrm{atom}}} (||\vec{x}_{i,\mathrm{pred}} - \vec{x}_{j,\mathrm{pred}}|| - ||\vec{x}_{i,\mathrm{gt}} - \vec{x}_{j,\mathrm{gt}}||)^2. \tag{B.7}$$

This quantity is invariant up to any rigid transformation $\mathcal{T} = (\mathcal{R}, \vec{t})$ of one of its inputs. Indeed,

$$
\begin{aligned}
\mathrm{PM}_{\mathrm{atom}}^2(\mathcal{T} \circ x_{\mathrm{pred}}, \vec{x}_{\mathrm{gt}}) &= \frac{1}{n_{\mathrm{atom}}^2} \sum_{i \in n_{\mathrm{atom}}} \sum_{j \in n_{\mathrm{atom}}} (||\mathcal{T} \circ \vec{x}_{i,\mathrm{pred}} - \mathcal{T} \circ \vec{x}_{j,\mathrm{pred}}|| - ||\vec{x}_{i,\mathrm{gt}} - \vec{x}_{j,\mathrm{gt}}||)^2 \\
&= \frac{1}{n_{\mathrm{atom}}^2} \sum_{i \in n_{\mathrm{atom}}} \sum_{j \in n_{\mathrm{atom}}} (||\mathcal{R}\vec{x}_{i,\mathrm{pred}} - \vec{t} - \mathcal{R}\vec{x}_{j,\mathrm{pred}} + \vec{t}|| - ||\vec{x}_{i,\mathrm{gt}} - \vec{x}_{j,\mathrm{gt}}||)^2 \\
&= \frac{1}{n_{\mathrm{atom}}^2} \sum_{i \in n_{\mathrm{atom}}} \sum_{j \in n_{\mathrm{atom}}} (||\mathcal{R}(\vec{x}_{i,\mathrm{pred}} - \vec{x}_{j,\mathrm{pred}})|| - ||\vec{x}_{i,\mathrm{gt}} - \vec{x}_{j,\mathrm{gt}}||)^2 \\
&= \frac{1}{n_{\mathrm{atom}}^2} \sum_{i \in n_{\mathrm{atom}}} \sum_{j \in n_{\mathrm{atom}}} (||\vec{x}_{i,\mathrm{pred}} - \vec{x}_{j,\mathrm{pred}}|| - ||\vec{x}_{i,\mathrm{gt}} - \vec{x}_{j,\mathrm{gt}}||)^2 \\
&= \mathrm{PM}_{\mathrm{atom}}^2(x^{\mathrm{pred}}, x^{\mathrm{gt}})
\end{aligned}
\tag{B.8}
$$

$\square$

**Theorem B.3.** *Geometric loss is SE3-invariant.*

*Proof.* We consider:

$$\mathcal{L}_{\mathrm{Geom}}(\mathcal{T}_{\mathrm{pred}}, \mathcal{T}_{\mathrm{gt}}) = \frac{1}{M-1} \sum_{i=1}^{M-1} \mathcal{L}_{\mathrm{RMSD}}(\mathcal{T}_{M,\mathrm{pred}}^{-1} \circ \mathcal{T}_{i,\mathrm{pred}}, \mathcal{T}_{M,\mathrm{gt}}^{-1} \circ \mathcal{T}_{i,\mathrm{gt}}). \tag{B.9}$$

This quantity is invariant up to any transformation $\mathcal{T}_{\mathrm{noise}}$ of one of its inputs:

$$
\begin{aligned}
\mathcal{L}_{\mathrm{Geom}}(\mathcal{T}_{\mathrm{noise}} \circ \mathcal{T}_{\mathrm{pred}}, \mathcal{T}_{\mathrm{gt}}) &= \frac{1}{M-1} \sum_{i=1}^{M-1} \mathcal{L}_{\mathrm{RMSD}}\Big( (\mathcal{T}_{\mathrm{noise}} \circ \mathcal{T}_{M,\mathrm{pred}})^{-1} \circ \mathcal{T}_{\mathrm{noise}} \circ \mathcal{T}_{i,\mathrm{pred}}, \\
&\qquad\qquad\qquad\qquad\qquad\qquad \mathcal{T}_{M,\mathrm{gt}}^{-1} \circ \mathcal{T}_{i,\mathrm{gt}} \Big) \\
&= \frac{1}{M-1} \sum_{i=1}^{M-1} \mathcal{L}_{\mathrm{RMSD}}\Big( \mathcal{T}_{M,\mathrm{pred}}^{-1} \circ \mathcal{T}_{\mathrm{noise}}^{-1} \circ \mathcal{T}_{\mathrm{noise}} \circ \mathcal{T}_{i,\mathrm{pred}}, \\
&\qquad\qquad\qquad\qquad\qquad\qquad \mathcal{T}_{M,\mathrm{gt}}^{-1} \circ \mathcal{T}_{i,\mathrm{gt}} \Big) \\
&= \frac{1}{M-1} \sum_{i=1}^{M-1} \mathcal{L}_{\mathrm{RMSD}}\Big( \mathcal{T}_{M,\mathrm{pred}}^{-1} \circ \mathcal{T}_{i,\mathrm{pred}}, \mathcal{T}_{M,\mathrm{gt}}^{-1} \circ \mathcal{T}_{i,\mathrm{gt}} \Big) \\
&= \mathcal{L}_{\mathrm{Geom}}(\mathcal{T}_{\mathrm{pred}}, \mathcal{T}_{\mathrm{gt}})
\end{aligned}
$$

$\square$

# C   Method and implementation

## C.1   Extension to the inversion dataset

We argue that our method can also be applied to the inversion version of the dataset. Indeed this version, defined in Liu et al. (2024b), presents half of the 17 molecules in each assembly as the left-handed and right-handed geometries of a chiral or achiral molecule. The latter molecules can interconvert during crystallization

and thus, our permutation-invariant approach can be applied on this dataset. In the case of chiral molecules which can not interconvert during crystallization, the invariance to permutation can be adapted to the 2 subsets of left-handed and right handed geometries individually.

## C.2 AssembleFlow atom-level model

We use the Atom-level implemented in AssembleFlow and which can be described in Algorithm C.1. It is composed of a PaiNN embedding layer to encode each molecular structure individually followed by $N$ layers of atom-to-molecules attention message passing. Each molecule's transformation prediction is then obtained by aggregating the resulting atomic embeddings per molecule and passed through a projection head.

---

**Algorithm C.1** Atom-level model.

---

**def** AtomModel($\{a_i\}$ : atoms, $t$ : time, $\{\vec{\mathbf{P}}_i^t\}$ : positions, $N_{\text{layer}} = 5, N_{\text{conv}} = 5, c = 128$)

1: $t = \text{Linear}(\text{SiLU}(\text{Linear}(\text{time\_embed}(t))))$        [c]

2: $\{h_i^t\} = \text{PaiNN}(\{a_i\}, \{\vec{\mathbf{P}}_i^t\}) + \text{Linear}(\text{SiLU}(t))$        $[N_{\text{atom}}, c]$

3: $\{s_i^t\} = \text{ScatterMean}_{\text{per mol}}(\{h_i^t\})$        $[N_{\text{mol}}, c]$

4: $\{\vec{\mathbf{X}}_i^t\} = \text{ScatterMean}_{\text{per mol}}(\{\vec{\mathbf{P}}_i^t\})$        $[N_{\text{mol}}, 3]$

5: $\{e_{ij}^t\} = \text{RadialGraph}(\{\vec{\mathbf{P}}_i^t\}, \{\vec{\mathbf{X}}_i^t\})$        Atom to Molecules edges

6: **for all** $\{i, j\}/e_{ij}^t = 1$ :

7:     $\Delta_{ij}^t = \vec{\mathbf{P}}_i^t - \vec{\mathbf{X}}_j^t/\|\vec{\mathbf{P}}_i^t - \vec{\mathbf{X}}_j^t\|$

8:     $\chi_{ij}^t = \vec{\mathbf{P}}_i^t \times \vec{\mathbf{X}}_j^t/\|\vec{\mathbf{P}}_i^t \times \vec{\mathbf{X}}_j^t\|$

9:     $\Lambda_{ij}^t = \Delta_{ij}^t \times \chi_{ij}^t$

10:    $\text{Base}_{ij}^t = \text{concat}(\Delta_{ij}^t, \chi_{ij}^t, \Lambda_{ij}^t)$        [Edges, 3, 3]

11:    $\mathbf{E}_i^t = \text{MLP}(\text{GaussianFourierEmbed}(\text{Base}_{ij}^t \cdot \vec{\mathbf{P}}_i^t))$        [Edges, c]

12:    $\mathbf{E}_j^t = \text{MLP}(\text{GaussianFourierEmbed}(\text{Base}_{ij}^t \cdot \vec{\mathbf{X}}_j^t))$        [Edges, c]

13:    $\{\mathbf{z}_{ij}^t\} = \text{MLP}(\text{concat}(\mathbf{E}_i^t, \mathbf{E}_j^t))$        [Edges, c]

14:**end for**

15:$\mathcal{R}_i^t = \mathbf{0}$ and $\mathcal{S}_i^t = \mathbf{0}$

16:**for all** $l \in [1, ..., N_{\text{layer}}]$:

17:    **for all** $f \in [1, ..., N_{\text{conv}}]$:

18:      $\{\tilde{\mathbf{h}}_i^t\} = \text{GAT}_{\text{conv}}^f(\{\mathbf{h}_i^t\}, \{\mathbf{s}_j^t\}, \{\mathbf{z}_{ij}^t\})$

18:      $\{\mathbf{h}_i^t\} = \{\mathbf{h}_i^t\} + \text{LayerNorm}(\{\tilde{\mathbf{h}}_i^t\})$

19:      $\{\tilde{\mathbf{h}}_i^t\} = \text{FFN}^f(\{\mathbf{h}_i^t\})$

20:      $\{\mathbf{h}_i^t\} = \{\mathbf{h}_i^t\} + \text{LayerNorm}(\{\tilde{\mathbf{h}}_i^t\}) + \text{Linear}(\text{SiLU}(t))$

21:      **if** $l < N_{\text{conv}}$ :

22:        $\{\mathbf{h}_i^t\} = \text{SiLU}(\{\mathbf{h}_i^t\})$

23:      **end if**

24:    **end for**

25:    $\{\mathbf{s}_i^t\} = \text{ScatterMean}_{\text{per mol}}(\{h_i^t\})$

26:    $\mathcal{R}_i^t \leftarrow \mathcal{R}_i^t + \text{ScatterMean}_{\text{per mol}}\Big($

         $\text{Mean}_{j \in \mathcal{N}(i)}\{\text{MLP}(\text{concat}(h_i^t + s_j^t, \mathbf{z}_{ij}^t)) \cdot \text{Base}_{ij}^t\}\Big)$        $[N_{mol}, 3]$

27:    $\mathcal{S}_i^t \leftarrow \mathcal{S}_i^t + \text{ScatterMean}_{\text{per mol}}\Big($

         $\text{Mean}_{j \in \mathcal{N}(i)}\{\text{Proj}\big(\text{Linear}\big(\text{MLP}(\text{concat}(h_i^t + s_j^t, \mathbf{z}_{ij}^t)) \cdot \text{Base}_{ij}^t\big)\big)\}\Big)$ $[N_{mol}, 4]$

28: **end for**

29: **return** $\{\mathcal{S}_i^t, \mathcal{R}_i^t\}$

---

### C.3 Implementation details

#### C.3.1 Hyperparameters and number of parameters

Table C.1 lists the hyperparameters used during training along with the number of parameters for the model and the memory usage.

Table C.1: **Hyperparameters used in the model.**

| Model part | Function | Parameters |
|---|---|---|
| Training | Epochs | {500} |
| | Batch size | {8} |
| | Loss | {LM: {alpha: 10}} {RMSD: ∅ } {Geometric: ∅ } |
| | Assignment | {None: ∅} {'Exact': ∅} {'Differentiable': {reg=$5.10^{-2}$.median_score}} |
| Optimizer | Name | {Adam} |
| | Learning rate | {$10^{-4}$} |
| | Weight decay | {0} |
| | Scheduler | {'CosineAnnealingLR'} |
| Molecular Encoder (PaiNN) | cutoff | {5} |
| | embedding dim | {128} |
| | number of interactions | {3} |
| | number of rbf | {20} |
| | scatter | {'mean'} |
| | gamma | {3.25} |
| Backbone (AssembleFlow Atom) | emb_dim | {128} |
| | hidden dim | {128} |
| | cutoff | {10} |
| | cluster cutoff | {50} |
| | number of timesteps | {1, 50} |
| | scatter | {'mean'} |
| | number of Gaussians | {20} |
| | number of heads | {8} |
| | number of layers | {5} |
| | number of convolutions | {5} |
| | gamma | {3.25} |
| **Total number of parameters:** | 4 292 718 | |
| **Total memory usage:** | 38.9 GB | |

#### C.3.2 Licenses and versions

The common environment packages are released with the code through a conda environment. We also report in Table C.2 the versions and licenses of the main packages used.

Table C.2: **Versions and licenses.**

| Package | Version | License |
|---|---|---|
| COD-Cluster17 | git commit | MIT |
| AssembleFlow Model | git commit | MIT |
| POT | 0.9.5 | MIT |
| RMSD | - | CeCILL |

## D  Ablation studies

### D.1  Differential assignment with direct regression

In Table D.1, we list the experiments of training or not with differential assignment in direct regression with the AssembleFlow atom-level model. We want to draw the attention to the PM* methods and the great added value of using our assignment method regardless of the loss being used.

Table D.1: Ablation study of using differentiable assignment (Diff. assign.) losses during training on COD-Cluster17 with direct regression.

| Loss | Diff. assign. | Test Loss in Å↓ | | Packing matching in Å↓ | | | |
|---|---|---|---|---|---|---|---|
| | | $\mathcal{L}^*_{\mathrm{RMSD}}$ | $\mathcal{L}^*_{\mathrm{Geom}}$ | $\mathrm{PM}^*_{\mathrm{center}}$ | $\mathrm{PM}^*_{\mathrm{atom}}$ | $\mathrm{PM}_{\mathrm{center}}$ | $\mathrm{PM}_{\mathrm{atom}}$ |
| Dataset: **COD-Cluster17-5K** | | | | | | | |
| $\mathcal{L}_{\mathrm{ML}}$ | | $9.64_{\pm 0.21}$ | $11.43_{\pm 0.08}$ | $5.62_{\pm 0.31}$ | $6.68_{\pm 0.24}$ | $6.97_{\pm 0.23}$ | $7.62_{\pm 0.18}$ |
| $\mathcal{L}_{\mathrm{RMSD}}$ | | $9.64_{\pm 0.03}$ | $11.24_{\pm 0.15}$ | $5.57_{\pm 0.19}$ | $6.67_{\pm 0.07}$ | $6.93_{\pm 0.12}$ | $7.61_{\pm 0.02}$ |
| $\mathcal{L}_{\mathrm{Geom}}$ | | $10.10_{\pm 0.14}$ | $10.05_{\pm 0.11}$ | $8.44_{\pm 0.43}$ | $8.37_{\pm 0.26}$ | $9.05_{\pm 0.37}$ | $8.74_{\pm 0.22}$ |
| $\mathcal{L}^*_{\mathrm{ML}}$ | ✓ | $8.69_{\pm 0.06}$ | $12.16_{\pm 0.12}$ | $3.60_{\pm 0.04}$ | $5.54_{\pm 0.04}$ | $5.80_{\pm 0.03}$ | $6.96_{\pm 0.03}$ |
| $\mathcal{L}^*_{\mathrm{RMSD}}$ | ✓ | $8.73_{\pm 0.07}$ | $12.05_{\pm 0.15}$ | $3.77_{\pm 0.12}$ | $5.67_{\pm 0.08}$ | $5.85_{\pm 0.05}$ | $6.98_{\pm 0.05}$ |
| $\mathcal{L}^*_{\mathrm{Geom}}$ | ✓ | $9.32_{\pm 0.06}$ | $8.78_{\pm 0.05}$ | $5.55_{\pm 0.15}$ | $6.54_{\pm 0.07}$ | $6.92_{\pm 0.07}$ | $7.46_{\pm 0.02}$ |
| Dataset: **COD-Cluster17-All** | | | | | | | |
| $\mathcal{L}_{\mathrm{ML}}$ | | $11.67_{\pm 0.07}$ | $11.33_{\pm 0.05}$ | $12.94_{\pm 0.16}$ | $10.47_{\pm 0.03}$ | $13.03_{\pm 0.15}$ | $10.47_{\pm 0.02}$ |
| $\mathcal{L}_{\mathrm{RMSD}}$ | | $11.58_{\pm 0.04}$ | $11.20_{\pm 0.12}$ | $12.98_{\pm 0.13}$ | $10.44_{\pm 0.01}$ | $13.07_{\pm 0.12}$ | $10.43_{\pm 0.01}$ |
| $\mathcal{L}_{\mathrm{Geom}}$ | | $11.90_{\pm 0.08}$ | $11.38_{\pm 0.09}$ | $13.62_{\pm 0.07}$ | $10.52_{\pm 0.01}$ | $13.66_{\pm 0.06}$ | $10.49_{\pm 0.01}$ |
| $\mathcal{L}^*_{\mathrm{ML}}$ | ✓ | $8.65_{\pm 0.02}$ | $12.10_{\pm 0.10}$ | $3.47_{\pm 0.04}$ | $5.51_{\pm 0.02}$ | $5.80_{\pm 0.00}$ | $7.00_{\pm 0.01}$ |
| $\mathcal{L}^*_{\mathrm{RMSD}}$ | ✓ | $8.70_{\pm 0.03}$ | $12.16_{\pm 0.08}$ | $3.41_{\pm 0.04}$ | $5.54_{\pm 0.01}$ | $5.80_{\pm 0.00}$ | $7.00_{\pm 0.01}$ |
| $\mathcal{L}^*_{\mathrm{Geom}}$ | ✓ | $9.35_{\pm 0.00}$ | $8.71_{\pm 0.03}$ | $5.43_{\pm 0.10}$ | $6.52_{\pm 0.05}$ | $6.84_{\pm 0.06}$ | $7.45_{\pm 0.02}$ |

### D.2  Differential assignment with flow matching

Table D.2 lists the experiments of switching on and off the expensive flow matching framework (table 3) along with using the differential assignment. The added value of flow matching when using the differential assignment loss is not very clear in the current framework. As it does not always significantly help the method, we suspect a need to further adapt the assignment method to the iterative flow matching scheme. However, we would like to point out two things. Firstly, it greatly improves the performance of the *relative* geometric method on the *absolute* metrics while decreasing it on the *relative* metric. Secondly, it enable to reach the overall best performance in the $\mathrm{PM}^*_{\mathrm{center}}$ metric.

We report in Table D.3 (resp. D.4) a clearer ablation study conducted with $\mathcal{L}_{\mathrm{RMSD}}$ (resp. $\mathcal{L}_{\mathrm{Geom}}$) of training the AssembleFlow backbone with or without flow matching and permutation-invariant loss.

Table D.2: Ablation study of using flow matching in addition to differentiable assignment losses during training on COD-Cluster17.

| Loss | Flow Matching | Test Loss in Å↓ | | Packing matching in Å↓ | | | |
|---|---|---|---|---|---|---|---|
| | | $\mathcal{L}^*_{\text{RMSD}}$ | $\mathcal{L}^*_{\text{Geom}}$ | $\text{PM}^*_{\text{center}}$ | $\text{PM}^*_{\text{atom}}$ | $\text{PM}_{\text{center}}$ | $\text{PM}_{\text{atom}}$ |
| Dataset: **COD-Cluster17-5K** | | | | | | | |
| $\mathcal{L}^*_{\text{ML}}$ | | $8.69_{\pm 0.06}$ | $12.16_{\pm 0.12}$ | $3.60_{\pm 0.04}$ | $5.54_{\pm 0.04}$ | $5.80_{\pm 0.03}$ | $6.96_{\pm 0.03}$ |
| $\mathcal{L}^*_{\text{RMSD}}$ | | $8.73_{\pm 0.07}$ | $12.05_{\pm 0.15}$ | $3.77_{\pm 0.12}$ | $5.67_{\pm 0.08}$ | $5.85_{\pm 0.05}$ | $6.98_{\pm 0.05}$ |
| $\mathcal{L}^*_{\text{Geom}}$ | | $9.32_{\pm 0.06}$ | $8.78_{\pm 0.05}$ | $5.55_{\pm 0.15}$ | $6.54_{\pm 0.07}$ | $6.92_{\pm 0.07}$ | $7.46_{\pm 0.02}$ |
| $\mathcal{L}^*_{\text{ML}}$ | ✓ | $9.31_{\pm 0.25}$ | $13.54_{\pm 0.50}$ | $3.48_{\pm 0.19}$ | $5.60_{\pm 0.14}$ | $6.12_{\pm 0.23}$ | $7.29_{\pm 0.21}$ |
| $\mathcal{L}^*_{\text{RMSD}}$ | ✓ | $9.53_{\pm 0.54}$ | $13.71_{\pm 0.40}$ | $3.43_{\pm 0.20}$ | $5.56_{\pm 0.14}$ | $6.12_{\pm 0.19}$ | $7.28_{\pm 0.17}$ |
| $\mathcal{L}^*_{\text{Geom}}$ | ✓ | $9.09_{\pm 0.09}$ | $10.48_{\pm 0.18}$ | $3.72_{\pm 0.11}$ | $5.73_{\pm 0.04}$ | $6.04_{\pm 0.10}$ | $7.19_{\pm 0.05}$ |
| Dataset: **COD-Cluster17-All** | | | | | | | |
| $\mathcal{L}^*_{\text{ML}}$ | | $8.65_{\pm 0.02}$ | $12.10_{\pm 0.10}$ | $3.47_{\pm 0.04}$ | $5.51_{\pm 0.02}$ | $5.80_{\pm 0.00}$ | $7.00_{\pm 0.01}$ |
| $\mathcal{L}^*_{\text{RMSD}}$ | | $8.70_{\pm 0.03}$ | $12.16_{\pm 0.08}$ | $3.41_{\pm 0.04}$ | $5.54_{\pm 0.01}$ | $5.80_{\pm 0.00}$ | $7.00_{\pm 0.01}$ |
| $\mathcal{L}^*_{\text{Geom}}$ | | $9.35_{\pm 0.00}$ | $8.71_{\pm 0.03}$ | $5.43_{\pm 0.10}$ | $6.52_{\pm 0.05}$ | $6.84_{\pm 0.06}$ | $7.45_{\pm 0.02}$ |
| $\mathcal{L}^*_{\text{ML}}$ | ✓ | $9.37_{\pm 0.09}$ | $13.69_{\pm 0.21}$ | $3.42_{\pm 0.12}$ | $5.63_{\pm 0.07}$ | $6.15_{\pm 0.12}$ | $7.36_{\pm 0.09}$ |
| $\mathcal{L}^*_{\text{RMSD}}$ | ✓ | $9.51_{\pm 0.38}$ | $13.42_{\pm 0.22}$ | $3.29_{\pm 0.04}$ | $5.53_{\pm 0.04}$ | $6.01_{\pm 0.06}$ | $7.23_{\pm 0.07}$ |
| $\mathcal{L}^*_{\text{Geom}}$ | ✓ | $9.28_{\pm 0.09}$ | $10.72_{\pm 0.13}$ | $3.89_{\pm 0.23}$ | $5.88_{\pm 0.12}$ | $6.27_{\pm 0.17}$ | $7.40_{\pm 0.12}$ |

Table D.3: Ablation study of using flow matching and differentiable assignment loss during training on COD-Cluster17-All with $\mathcal{L}_{\text{RMSD}}$.

| Flow Matching | Permutation Invariance | $\mathcal{L}^*_{\text{RMSD}}$ | $\text{PM}^*_{\text{center}}$ | $\text{PM}^*_{\text{atom}}$ |
|---|---|---|---|---|
| | | $11.58_{\pm 0.04}$ | $12.98_{\pm 0.13}$ | $10.44_{\pm 0.01}$ |
| ✓ | | $9.08_{\pm 0.12}$ | $3.51_{\pm 0.04}$ | $5.60_{\pm 0.03}$ |
| | ✓ | $8.70_{\pm 0.03}$ | $3.41_{\pm 0.04}$ | $5.54_{\pm 0.01}$ |
| ✓ | ✓ | $9.51_{\pm 0.38}$ | $3.29_{\pm 0.04}$ | $5.53_{\pm 0.04}$ |

Table D.4: Ablation study of using flow matching and differentiable assignment loss during training on COD-Cluster17-All with $\mathcal{L}_{\text{Geom}}$.

| Flow Matching | Permutation Invariance | $\mathcal{L}^*_{\text{Geom}}$ | $\text{PM}^*_{\text{center}}$ | $\text{PM}^*_{\text{atom}}$ |
|---|---|---|---|---|
| | | $11.38_{\pm 0.09}$ | $13.62_{\pm 0.07}$ | $10.52_{\pm 0.01}$ |
| ✓ | | $10.77_{\pm 0.13}$ | $3.78_{\pm 0.09}$ | $5.76_{\pm 0.05}$ |
| | ✓ | $8.71_{\pm 0.03}$ | $5.43_{\pm 0.10}$ | $6.52_{\pm 0.05}$ |
| ✓ | ✓ | $10.72_{\pm 0.13}$ | $3.89_{\pm 0.23}$ | $5.88_{\pm 0.12}$ |

Table D.5: Baseline study comparing the performance of the AssembleFlow model in the single molecular knowledge direct regression training scheme.

| Loss | Method | Test Loss in Å↓ | | Packing matching in Å↓ | | | |
|------|--------|--------------------------|--------------------------|------------------------------|------------------------------|------------------------------|------------------------------|
| | | $\mathcal{L}^*_{\text{RMSD}}$ | $\mathcal{L}^*_{\text{Geom}}$ | $\text{PM}^*_{\text{center}}$ | $\text{PM}^*_{\text{atom}}$ | $\text{PM}_{\text{center}}$ | $\text{PM}_{\text{atom}}$ |
| Dataset: **COD-Cluster17-5K** | | | | | | | |
| $\mathcal{L}_{\text{ML}}$ | AssembleFlow | $9.56_{\pm 0.07}$ | $13.35_{\pm 0.30}$ | $3.86_{\pm 0.13}$ | $5.79_{\pm 0.12}$ | $6.28_{\pm 0.15}$ | $7.39_{\pm 0.16}$ |
| $\mathcal{L}_{\text{RMSD}}$ | AssembleFlow | $9.41_{\pm 0.19}$ | $13.37_{\pm 0.24}$ | $3.89_{\pm 0.16}$ | $5.80_{\pm 0.16}$ | $6.26_{\pm 0.20}$ | $7.35_{\pm 0.21}$ |
| $\mathcal{L}_{\text{Geom}}$ | AssembleFlow | $9.22_{\pm 0.06}$ | $10.45_{\pm 0.55}$ | $3.90_{\pm 0.08}$ | $5.85_{\pm 0.04}$ | $6.14_{\pm 0.02}$ | $7.20_{\pm 0.05}$ |
| $\mathcal{L}^*_{\text{ML}}$ | SinkFast | $\mathbf{8.69_{\pm 0.06}}$ | $12.16_{\pm 0.12}$ | $\mathbf{3.60_{\pm 0.04}}$ | $\mathbf{5.54_{\pm 0.04}}$ | $\mathbf{5.80_{\pm 0.03}}$ | $\mathbf{6.96_{\pm 0.03}}$ |
| $\mathcal{L}^*_{\text{RMSD}}$ | SinkFast | $\mathbf{8.73_{\pm 0.07}}$ | $12.05_{\pm 0.15}$ | $3.77_{\pm 0.12}$ | $5.67_{\pm 0.08}$ | $\mathbf{5.85_{\pm 0.05}}$ | $\mathbf{6.98_{\pm 0.05}}$ |
| $\mathcal{L}^*_{\text{Geom}}$ | SinkFast | $9.32_{\pm 0.06}$ | $8.78_{\pm 0.05}$ | $5.55_{\pm 0.15}$ | $6.54_{\pm 0.07}$ | $6.92_{\pm 0.07}$ | $7.46_{\pm 0.02}$ |
| $\mathcal{L}_{\text{ML}}$ | Single Mol. | $11.35_{\pm 0.09}$ | $10.88_{\pm 0.07}$ | $12.18_{\pm 0.11}$ | $10.16_{\pm 0.06}$ | $12.26_{\pm 0.11}$ | $10.17_{\pm 0.06}$ |
| $\mathcal{L}_{\text{RMSD}}$ | Single Mol. | $11.35_{\pm 0.09}$ | $10.92_{\pm 0.08}$ | $12.46_{\pm 0.17}$ | $10.22_{\pm 0.07}$ | $12.53_{\pm 0.17}$ | $10.22_{\pm 0.06}$ |
| $\mathcal{L}_{\text{Geom}}$ | Single Mol. | $11.32_{\pm 0.12}$ | $10.87_{\pm 0.12}$ | $12.43_{\pm 0.24}$ | $10.20_{\pm 0.09}$ | $12.50_{\pm 0.23}$ | $10.20_{\pm 0.08}$ |
| $\mathcal{L}^*_{\text{ML}}$ | Single Mol. | $9.02_{\pm 0.11}$ | $12.34_{\pm 0.22}$ | $4.94_{\pm 0.07}$ | $6.35_{\pm 0.05}$ | $6.78_{\pm 0.08}$ | $7.59_{\pm 0.09}$ |
| $\mathcal{L}^*_{\text{RMSD}}$ | Single Mol. | $9.05_{\pm 0.05}$ | $12.58_{\pm 0.11}$ | $5.02_{\pm 0.17}$ | $6.42_{\pm 0.10}$ | $6.94_{\pm 0.12}$ | $7.72_{\pm 0.07}$ |
| $\mathcal{L}^*_{\text{Geom}}$ | Single Mol. | $9.06_{\pm 0.07}$ | $\mathbf{8.15_{\pm 0.03}}$ | $5.02_{\pm 0.07}$ | $6.44_{\pm 0.03}$ | $6.61_{\pm 0.07}$ | $7.48_{\pm 0.03}$ |

### D.3 Direction Regression with a single molecule knowledge

Another additional baseline experiment can be to directly predict the set of 17 rigid transformations from a single molecule, without knowledge of the set of 17 initial positions. We report the results in the Table D.5. We refer to this baseline as the Single Mol. method. As expected, it underperforms both SinkFast and AssembleFlow, since the AssembleFlow-Atom model specifically leverages intra-molecular interactions. This baseline supports the importance of incorporating such information.

### D.4 Using linear sum assignment during training against differentiable assignment

We report in Table D.6 the experiment of using the linear sum assignment (*exact*) during training against the differential assignment (*relaxed*). On the one hand, using the exact solver during training enables backpropagation for each molecule in the assembly along the path leading to its assigned target, while killing the other gradients corresponding to other paths to unassigned targets. On the other hand, the relaxed differential version preserves the gradients to all possible paths with probability attached to each, which enables a more diverse learning. While being suboptimal compared to the differential assignment, the added value of using the latter is very small as shown in Table D.6. We report here the performance obtained without tuning the regularization parameter of the Sinkhorn algorithm and exploring its influence on the overall performance. Nonetheless we argue that this hyperparameter should should play an important role with better-performing methods in the future. Indeed we believe that if the method learned nearly perfectly to match a molecule to its target position, this relaxed method would diversify the search space and act as a data augmentation method, the amount of which would be set by the regularization parameter.

### D.5 Angular VS translational prediction

We report in table D.7 the decomposition of the RMSD score in both its translation $\mathcal{L}^*_{\mathbb{R}^3}$ and rotation $\mathcal{L}^*_{\text{SO}(3)}$ parts. Please note that ${\mathcal{L}^*_{\text{RMSD}}}^2 = {\mathcal{L}^*_{\mathbb{R}^3}}^2 + {\mathcal{L}^*_{\text{SO}(3)}}^2$, following eq. 8. The noise baseline is computed by always using an identity transformation as a prediction, meaning, a zero translation and an identity rotation, and computing the RMSD between the sets $\mathcal{S}_{\text{initial}}$ of initial positions and $\mathcal{S}_{\text{final}}$ of final positions. Presented results indicate the scale of the problem and show in particular that initial orientations are better than predicted ones. This table shows that both models mainly focus on the translation part of the problem, while discarding rotations completely.

Table D.6: Ablation study of using differential or exact assignment losses during training on COD-Cluster17 with direct regression.

| Loss | Assignment type | Test Loss in Å↓ | | Packing matching in Å↓ | | | |
| | | $\mathcal{L}^*_{\text{RMSD}}$ | $\mathcal{L}^*_{\text{Geom}}$ | $\text{PM}^*_{\text{center}}$ | $\text{PM}^*_{\text{atom}}$ | $\text{PM}_{\text{center}}$ | $\text{PM}_{\text{atom}}$ |
|---|---|---|---|---|---|---|---|
| | | Dataset: **COD-Cluster17-5K** | | | | | |
| $\mathcal{L}^*_{\text{ML}}$ | Exact | $8.70_{\pm 0.06}$ | $12.24_{\pm 0.14}$ | $3.64_{\pm 0.12}$ | $5.56_{\pm 0.08}$ | $5.81_{\pm 0.04}$ | $6.96_{\pm 0.04}$ |
| $\mathcal{L}^*_{\text{RMSD}}$ | Exact | $8.72_{\pm 0.07}$ | $12.19_{\pm 0.05}$ | $3.65_{\pm 0.05}$ | $5.61_{\pm 0.03}$ | $5.81_{\pm 0.02}$ | $6.96_{\pm 0.04}$ |
| $\mathcal{L}^*_{\text{Geom}}$ | Exact | $9.32_{\pm 0.05}$ | $8.80_{\pm 0.08}$ | $5.51_{\pm 0.25}$ | $6.53_{\pm 0.14}$ | $6.90_{\pm 0.14}$ | $7.45_{\pm 0.06}$ |
| $\mathcal{L}^*_{\text{ML}}$ | Diff. | $8.69_{\pm 0.06}$ | $12.16_{\pm 0.12}$ | $3.60_{\pm 0.04}$ | $5.54_{\pm 0.04}$ | $5.80_{\pm 0.03}$ | $6.96_{\pm 0.03}$ |
| $\mathcal{L}^*_{\text{RMSD}}$ | Diff. | $8.73_{\pm 0.07}$ | $12.05_{\pm 0.15}$ | $3.77_{\pm 0.12}$ | $5.67_{\pm 0.08}$ | $5.85_{\pm 0.05}$ | $6.98_{\pm 0.05}$ |
| $\mathcal{L}^*_{\text{Geom}}$ | Diff. | $9.32_{\pm 0.06}$ | $8.78_{\pm 0.05}$ | $5.55_{\pm 0.15}$ | $6.54_{\pm 0.07}$ | $6.92_{\pm 0.07}$ | $7.46_{\pm 0.02}$ |
| | | Dataset: **COD-Cluster17-All** | | | | | |
| $\mathcal{L}^*_{\text{ML}}$ | Exact | $8.65_{\pm 0.02}$ | $12.18_{\pm 0.02}$ | $3.37_{\pm 0.03}$ | $5.47_{\pm 0.02}$ | $5.78_{\pm 0.01}$ | $6.99_{\pm 0.01}$ |
| $\mathcal{L}^*_{\text{RMSD}}$ | Exact | $8.70_{\pm 0.03}$ | $12.14_{\pm 0.09}$ | $3.44_{\pm 0.09}$ | $5.56_{\pm 0.03}$ | $5.80_{\pm 0.01}$ | $7.00_{\pm 0.01}$ |
| $\mathcal{L}^*_{\text{Geom}}$ | Exact | $9.35_{\pm 0.03}$ | $8.71_{\pm 0.03}$ | $5.40_{\pm 0.07}$ | $6.51_{\pm 0.05}$ | $6.84_{\pm 0.05}$ | $7.46_{\pm 0.03}$ |
| $\mathcal{L}^*_{\text{ML}}$ | Diff. | $8.65_{\pm 0.02}$ | $12.10_{\pm 0.10}$ | $3.47_{\pm 0.04}$ | $5.51_{\pm 0.02}$ | $5.80_{\pm 0.00}$ | $7.00_{\pm 0.01}$ |
| $\mathcal{L}^*_{\text{RMSD}}$ | Diff. | $8.70_{\pm 0.03}$ | $12.16_{\pm 0.08}$ | $3.41_{\pm 0.04}$ | $5.54_{\pm 0.01}$ | $5.80_{\pm 0.00}$ | $7.00_{\pm 0.01}$ |
| $\mathcal{L}^*_{\text{Geom}}$ | Diff. | $9.35_{\pm 0.00}$ | $8.71_{\pm 0.03}$ | $5.43_{\pm 0.10}$ | $6.52_{\pm 0.05}$ | $6.84_{\pm 0.06}$ | $7.45_{\pm 0.02}$ |

Table D.7: Both AssembleFlow and SinkFast RMSD performance decomposed between translation and rotation on COD-Cluster17-5K. Baseline scores indicate the scale of the metric and are computed between $\mathcal{S}_{\text{initial}}$ and $\mathcal{S}_{\text{final}}$ as if the model always predicts identity transformations.

| Model | Loss | Test Loss in Å↓ | | |
| | | $\mathcal{L}^*_{\text{RMSD}}$ | $\mathcal{L}^*_{\mathbb{R}^3}$ | $\mathcal{L}^*_{\text{SO}(3)}$ |
|---|---|---|---|---|
| | | Dataset: **COD-Cluster17-5K** | | |
| Noise (Baseline) | | $12.83_{\pm 0.05}$ | $11.55_{\pm 0.02}$ | $5.42_{\pm 0.06}$ |
| AssembleFlow | $\mathcal{L}_{\text{ML}}$ | $9.53_{\pm 0.09}$ | $7.60_{\pm 0.09}$ | $5.65_{\pm 0.06}$ |
| | $\mathcal{L}_{\text{RMSD}}$ | $9.43_{\pm 0.23}$ | $7.47_{\pm 0.21}$ | $5.66_{\pm 0.10}$ |
| | $\mathcal{L}_{\text{Geom}}$ | $9.12_{\pm 0.05}$ | $7.10_{\pm 0.07}$ | $5.65_{\pm 0.14}$ |
| SinkFast | $\mathcal{L}^*_{\text{ML}}$ | $8.90_{\pm 0.11}$ | $6.63_{\pm 0.06}$ | $5.87_{\pm 0.09}$ |
| | $\mathcal{L}^*_{\text{RMSD}}$ | $8.86_{\pm 0.09}$ | $6.66_{\pm 0.06}$ | $5.77_{\pm 0.07}$ |
| | $\mathcal{L}^*_{\text{Geom}}$ | $9.33_{\pm 0.11}$ | $7.49_{\pm 0.09}$ | $5.50_{\pm 0.07}$ |

**Equivariant version.** We have made an attempt to make the output of the model equivariant with the aim to improve in particular the orientation predictions. The implementation of this adaptation is presented in Algorithm D.1. We report the results of the experiment in Table D.8. Unfortunately, the problem remains unsolved and we believe that the model definition requires deeper modifications. It requires more advanced equivariant techniques to improve the expressivity of its intermediate layers, which results in changing completely the AssembleFlow backbone.

---

**Algorithm D.1** Equivariant atom-level model.

---

$\mathbf{def}$ AtomModel($\{f_a\}$ : atomic features, $t$ : time, $\{\vec{\mathbf{P}}_a^t\}$ : atomic positions,
$\qquad\qquad \{\mathcal{Q}_m^t\}$ : molecular orientations, $N_{\text{layer}} = 5, N_{\text{conv}} = 5, c = 128$)

1 : $t = \text{MLP}(\text{time\_embed}(t))$ $\hfill [c]$
2 : $\{h_a^t\} = \text{PaiNN}(\{f_a\}, \{\vec{\mathbf{P}}_a^t\}) + \text{Linear}(\text{SiLU}(t))$ $\hfill [N_{\text{atom}}, c]$
3 : $\{s_m^t\} = \text{ScatterMean}_{\text{per mol}}(\{h_a^t\})$ $\hfill [N_{\text{mol}}, c]$
4 : $\{\vec{\mathbf{X}}_m^t\} = \text{ScatterMean}_{\text{per mol}}(\{\vec{\mathbf{P}}_a^t\})$ $\hfill [N_{\text{mol}}, 3]$
5 : $\{e_{ij}^t\} = \text{RadialGraph}(\{\vec{\mathbf{P}}_a^t\}, \{\vec{\mathbf{X}}_m^t\})$ $\hfill$ Atom to Molecules edges
6 : $\mathbf{for\ all}\ \{i,j\}/e_{ij}^t = 1$ : $\hfill$ Each molecule j that is the neighbor of each atom i
7 : $\quad \Delta_{ij}^t, \chi_{ij}^t, \Lambda_{ij}^t = \text{OrthNorm}(\vec{\mathbf{P}}_{m,0}^t, \vec{\mathbf{P}}_{m,2}^t, \vec{\mathbf{P}}_{m,3}^t)$ 3 first atoms of molecule m s.t. $i \in m$.
8 : $\quad \text{Base}_{ij}^t = \text{concat}(\Delta_{ij}^t, \chi_{ij}^t, \Lambda_{ij}^t) \mid \text{origin: } \vec{O}_{ij}^t = \vec{P}_{m,0}^t$ $\hfill [\text{Edges, 3, 3}]$
9 : $\quad \mathbf{E}_i^t = \text{MLP}(\text{GaussianFourierEmbed}(\text{Base}_{ij}^t \cdot [\vec{\mathbf{P}}_i^t - \vec{O}_{ij}^t]))$ $\hfill [\text{Edges, c}]$
10: $\quad \mathbf{E}_j^t = \text{MLP}(\text{GaussianFourierEmbed}(\text{Base}_{ij}^t \cdot [\vec{\mathbf{X}}_j^t - \vec{O}_{ij}^t]))$ $\hfill [\text{Edges, c}]$
11: $\quad \{\mathbf{z}_{ij}^t\} = \text{MLP}(\text{concat}(\mathbf{E}_i^t, \mathbf{E}_j^t))$ $\hfill [\text{Edges, c}]$
12: $\mathbf{end\ for}$
13: $\mathcal{R}_m^t = \vec{\mathbf{P}}_m^t$ and $\mathcal{S}_m^t = \text{Base}_{m,j=0}^t$ as quaternion
14: $\mathcal{A}_i^t = \mathcal{S}_m^t \quad \forall m \le N_{\text{mol}}$ s.t. $i \in$ molecule m.
15: $\mathbf{for\ all}\ l \in [1, ..., N_{\text{layer}}]$:
16: $\quad \mathbf{for\ all}\ f \in [1, ..., N_{\text{conv}}]$:
17: $\quad\quad \{\tilde{\mathbf{h}}_i^t\} = \text{GAT}_{\text{conv}}^f(\{\mathbf{h}_i^t\}, \{\mathbf{s}_j^t\}, \{\text{concat}(\mathbf{z}_{ij}^t, \mathcal{A}_i^t(\mathcal{Q}_j^t)^{-1}, (\text{Base}_{ij}^t)^{-1}(\mathbf{P}_i^t - \mathbf{X}_j^t))\})$
18: $\quad\quad \{\mathbf{h}_i^t\} = \{\mathbf{h}_i^t\} + \text{LayerNorm}(\{\tilde{\mathbf{h}}_i^t\})$
19: $\quad\quad \{\tilde{\mathbf{h}}_i^t\} = \text{FFN}^f(\{\mathbf{h}_i^t\})$
20: $\quad\quad \{\mathbf{h}_i^t\} = \{\mathbf{h}_i^t\} + \text{LayerNorm}(\{\tilde{\mathbf{h}}_i^t\}) + \text{Linear}(\text{SiLU}(t))$
21: $\quad\quad \mathbf{if}\ l < N_{\text{conv}}$ :
22: $\quad\quad\quad \{\mathbf{h}_i^t\} = \text{SiLU}(\{\mathbf{h}_i^t\})$
23: $\quad\quad \mathbf{end\ if}$
24: $\quad \mathbf{end\ for}$
25: $\quad \{\mathbf{s}_m^t\} = \text{Mean}_{i \in m}(\{h_i^t\})$
26: $\quad \{\mathcal{F}_{ij}^t\} = \text{concat}(h_i^t + s_j^t, \mathbf{z}_{ij}^t, \mathcal{A}_i^t(\mathcal{Q}_j^t)^{-1}, (\text{Base}_{ij}^t)^{-1}(\mathbf{P}_i^t - \mathbf{X}_j^t))$ $\hfill [\text{Edges, 2*c+7}]$
27: $\quad \mathcal{R}_m^t \leftarrow \mathcal{R}_m^t + \text{Mean}_{i \in m}\left(\text{Mean}_{j \in \mathcal{N}(i)}\{\text{MLP}(\mathcal{F}_{ij}^t) \cdot \text{Base}_{ij}^t\}\right)$ $\hfill [N_{\text{mol}}, 3]$
28: $\quad \mathcal{A}_i^t \leftarrow \text{Mean}_{j \in \mathcal{N}(i)}\left(\text{Proj}\left(\text{Linear}\left(\text{MLP}(\mathcal{F}_{ij}^t) \cdot \text{Base}_{ij}^t\right)\right)\right).\mathcal{A}_i^t$ $\hfill [N_{\text{mol}}, 4]$
29: $\quad \mathcal{S}_m^t \leftarrow \text{RotationMean}_{i \in m}(\mathcal{S}_m^t, \mathcal{A}_i^t, h_i^t)$ $\hfill [N_{\text{mol}}, 4]$
30: $\mathbf{end\ for}$
31: $\mathbf{return}\ \{\mathcal{S}_i^t, \mathcal{R}_i^t\}$

---

### D.6 Dependency to Skinhorn regularization parameter

In Table D.9, we report the experiments of training our SinkFast model with different values of the regularization coefficient in the Sinkhorn algorithm. We can observe that the model is quite sensitive to this parameter with the best results obtained with a regularization value of 10. For such a value of this parameter, the permutation probability matrix is smooth, neither too uniform nor too sharp.

---

**Algorithm D.2** Aggregation function of quaternions.

---

**def** RotationMean$_{\text{index}}$($\{\mathbf{q}_{1,i}\}$ : initial quaternions, $\{\mathbf{q}_{2,j}\}$ : quaternions to aggregate,
$\{\mathcal{E}_{ij}\}$ : edge attribute between i and j, $\lambda = 1$ : regularization)

2: $\{w_{ij}\} = \text{Linear}(\{\mathcal{E}_{ij}\})$          [Edges, 1]

3: $\mathcal{M}_i = \dfrac{\sum_{j \in \mathcal{N}(i)} w_{ij}^2 \mathbf{q}_{2,j}^{-1} \times \mathbf{q}_{2,j}^{-1}}{\sum_{j \in \mathcal{N}(i)} w_{ij}^2}$    with $\times$ the outer product $[N_1, 4, 4]$

4: $\mathcal{M}^{\text{Sym}} = \frac{1}{2}(\mathcal{M} + \mathcal{M}^T)$          $[N_1, 4, 4]$

3: $U_i = \text{DomEigVec}(\mathcal{M}_i^{\text{Sym}} + \lambda.\mathbf{q}_{1,i}^{-1} \times \mathbf{q}_{1,i}^{-1})$    get dominant eigen vector $[N_1, 4]$

5: $\mathbf{q}_{1,i} \leftarrow \mathbf{q}_{1,i}.U_i.\mathbf{q}_{1,i}$

6: **return** $\{\mathbf{q}_{1,i}\}$

---

Table D.8: Ablation study of making SinkFast's backbone equivariant and trained with flow matching on COD-Clutser17 - 5k. The model is trained with $L_{\text{Geom}}$ loss as it is then purposed for relative prediction tasks.

| Flow Matching | Equivariant Backbone | $\mathcal{L}_{\text{Geom}}^*$ | $\text{PM}_{\text{center}}^*$ | $\text{PM}_{\text{atom}}^*$ |
|:---:|:---:|:---:|:---:|:---:|
|  |  | 8.87 | 5.48 | 6.53 |
|  | ✓ | 9.22 | 5.55 | 6.66 |
| ✓ |  | 10.34 | 3.99 | 5.88 |
| ✓ | ✓ | 10.26 | 4.79 | 6.27 |

Table D.9: Sensitivity experiment of SinkFast method to Sinkhorn's regularization coefficient. The model is trained with $L_{\text{RMSD}}^*$ loss on COD-Clutser-5k.

| Sinkhorn Regularization Parameter | $\mathcal{L}_{\text{RMSD}}^*$ | $\text{PM}_{\text{center}}^*$ | $\text{PM}_{\text{atom}}^*$ |
|:---:|:---:|:---:|:---:|
| 100 | $9.56_{\pm 0.07}$ | $6.58_{\pm 0.21}$ | $7.27_{\pm 0.07}$ |
| 50 | $8.95_{\pm 0.06}$ | $3.73_{\pm 0.08}$ | $5.73_{\pm 0.07}$ |
| 10 | $8.87_{\pm 0.05}$ | $3.64_{\pm 0.06}$ | $5.66_{\pm 0.05}$ |
| 5 | $8.89_{\pm 0.05}$ | $3.76_{\pm 0.04}$ | $5.72_{\pm 0.04}$ |
| 2 | $8.89_{\pm 0.10}$ | $3.87_{\pm 0.13}$ | $5.79_{\pm 0.10}$ |
| 1 | $8.91_{\pm 0.09}$ | $3.99_{\pm 0.05}$ | $5.85_{\pm 0.04}$ |
| 1e-1 | $8.92_{\pm 0.08}$ | $4.11_{\pm 0.04}$ | $5.91_{\pm 0.05}$ |
| 1e-2 | $8.99_{\pm 0.02}$ | $4.07_{\pm 0.00}$ | $5.90_{\pm 0.02}$ |
| 1e-3 | $9.12_{\pm 0.07}$ | $4.50_{\pm 0.03}$ | $6.09_{\pm 0.03}$ |

# E  Additional experiments

## E.1  OMC25-Cluster17 dataset

COD-Cluster17 is a molecular assembly dataset extracted from the Crystallography Open Database using the procedure described in Section 3. Following the same procedure, we constructed an additional dataset, OMC25-Cluster17, from the OMC25 database to compare our method with the state of the art.

Although OMC25 contains 25 million crystal structures, many are redundant. The authors start the OE62 dataset (Stuke et al., 2020) for each crystal of which they sample about 120 initial configurations and predict new arrangements for these crystals in different space groups. This produces final structures that are extremely similar. To avoid multiple data–target pairs with identical inputs but different targets—which would mislead training and introduce heavy redundancy—we filtered these down to approximately 200k distinct structures. Moreover, each molecule appears in about five different space-group configurations. To ensure a single target per data point, we retained only the configuration with minimum potential energy, resulting in about 43k distinct samples. While this significantly reduces the dataset size, it provides a more coherent learning setup.

Including all 120 similar structures would require a carefully controlled train/validation/test split, effectively serving as data augmentation. While this might slightly improve performance, it would not change the conclusions. Including all five space-group configurations would require a model explicitly conditioned on the space group, which neither SinkFast nor other baseline methods used in this study support, and is therefore beyond the scope of this study.

Table 2 presents the results. The Baseline corresponds to leaving molecules in their initial state and provides a reference scale. The conclusions on OMC25 mirror those on COD: SinkFast outperforms AssembleFlow or performs comparably, without relying on a flow-matching training scheme.

## E.2  Comparison to inorganic-based methods

### E.2.1  Training models from scratch

Inorganic crystal structure prediction is a fast-moving domain in which many state of the art models compete and innovate. We here want to compare the performance of current organic state of the art to the inorganic one. Thus, we conduct experiments on the COD-Cluster17-5k dataset by retraining both CDVAE (Xie et al., 2022) and DiffCSP (Jiao et al., 2023) models. In both cases, the models are trained to predict the target set of atomic positions from a noise distribution, where the same atoms are randomly positioned in space. Both methods operate in fractional coordinates and require a lattice definition. However, since the COD-Cluster17 dataset provides only point clouds without explicit lattice parameters or periodic boundary conditions, we define a pseudo lattice as the bounding box that encompasses all sets of molecules. Atom positions are then expressed in fractional coordinates relative to this pseudo lattice.

This setup introduces a stringent constraint that is not optimal for symmetry-based algorithms like CDVAE and DiffCSP, as we do not supply accurate information about atomic density or minimal symmetry groups. Despite this, both methods were able to produce high-quality predictions in certain cases. Notably, their performance did not show a strong correlation with the number of atoms per ASU.

At inference, we sample from the learned distribution of atomic positions rather than using initial positions provided by COD-Cluster17. As shown in Table E.1, both CDVAE and DiffCSP underperform significantly compared to rigid-body-based AssembleFlow and SinkFast methods, indicating that these point cloud models are not well suited to this task out-of-the-box. In Tables E.2 and E.3 we explore whether these methods perform particularly well on small graphs, but this tendency is actually also shared by both AssembleFlow and SinkFast.

We present in Table E.4 for each model the best predictions based on minimal Packing Matching (PM) score, and in Tables E.5 and E.6 the 5th and 10th percentiles, respectively. However, due to CDVAE's long training and very slow inference time, we compute its performance on 120 test samples. To ensure a fair comparison, we evaluate all models on this shared subset, which we refer to as the CDVAE subset. We observe from these experiments that CDVAE and DiffCSP can perform extremely well on very few structures. However, their

Table E.1: Performance in Å($\downarrow$) of our proposed SinkFast and AssembleFlow rigid-body methods against inorganic crystal structure prediction models CDVAE and DiffCSP on COD_Cluster17 - 5k test set.

| Method | $\mathbf{PM}^*_{\mathbf{center}}$ | $\mathbf{PM}^*_{\mathbf{atom}}$ |
|---|---|---|
| CDVAE | $10.50_{\pm 0.52}$ | $14.81_{\pm 0.89}$ |
| DiffCSP | $23.50_{\pm 2.44}$ | $30.61_{\pm 2.53}$ |
| AssembleFlow | $3.76_{\pm 0.00}$ | $5.73_{\pm 0.02}$ |
| SinkFast | $3.60_{\pm 0.04}$ | $5.54_{\pm 0.04}$ |

Table E.2: Performance in Å($\downarrow$) of our proposed SinkFast and AssembleFlow rigid-body methods against inorganic crystal structure prediction models CDVAE and DiffCSP on COD_Cluster17 - 5k test set filtered on $n_{\mathrm{atom}} \leq 16$ corresponding to the 20 smallest graphs.

| Method | $\mathbf{PM}^*_{\mathbf{center}}$ | $\mathbf{PM}^*_{\mathbf{atom}}$ |
|---|---|---|
| CDVAE | $8.17_{\pm 0.07}$ | $12.34_{\pm 0.91}$ |
| DiffCSP | $19.74_{\pm 0.42}$ | $25.89_{\pm 0.48}$ |
| AssembleFlow | $2.58_{\pm 0.19}$ | $3.49_{\pm 0.19}$ |
| SinkFast | $2.60_{\pm 0.04}$ | $3.48_{\pm 0.11}$ |

Table E.3: Performance in Å($\downarrow$) of our proposed SinkFast and AssembleFlow rigid-body methods against inorganic crystal structure prediction models CDVAE and DiffCSP on COD_Cluster17 - 5k test set filtered on $n_{\mathrm{atom}} \leq 50$ corresponding to half of the dataset.

| Method | $\mathbf{PM}^*_{\mathbf{center}}$ | $\mathbf{PM}^*_{\mathbf{atom}}$ |
|---|---|---|
| CDVAE | $10.37_{\pm 0.82}$ | $14.63_{\pm 1.10}$ |
| DiffCSP | $22.93_{\pm 2.66}$ | $29.98_{\pm 2.91}$ |
| AssembleFlow | $3.26_{\pm 0.06}$ | $4.96_{\pm 0.03}$ |
| SinkFast | $3.35_{\pm 0.11}$ | $4.95_{\pm 0.06}$ |

effectiveness quickly decreases across the dataset. This suggests that while these models have potential, they require further adaptation to be competitive on this task. In our view, adapting such models meaningfully goes beyond a quick out-of-the-box comparison. Nonetheless, they represent promising directions and could enrich the set of baselines on COD-Cluster17 in future dedicated studies or reviews.

Table E.4: Single best structure performance in Å($\downarrow$) of our proposed SinkFast and AssembleFlow rigid-body methods against inorganic crystal structure prediction models CDVAE and DiffCSP on COD_Cluster17 - 5k test set : filtered on the CDVAE subset.

| Method | $\mathbf{PM}^*_{\mathbf{center}}$ | $\mathbf{PM}^*_{\mathbf{atom}}$ |
| --- | --- | --- |
| CDVAE | 1.19 | 2.57 |
| DiffCSP | 0.99 | 4.61 |
| AssembleFlow | 2.04 | 3.03 |
| SinkFast | 2.06 | 2.73 |

Table E.5: 5th percentile performance in Å($\downarrow$) of our proposed SinkFast and AssembleFlow rigid-body methods against inorganic crystal structure prediction models CDVAE and DiffCSP on COD_Cluster17 - 5k test set : filtered on the CDVAE subset.

| Method | $\mathbf{PM}^*_{\mathbf{center}}$ | $\mathbf{PM}^*_{\mathbf{atom}}$ |
| --- | --- | --- |
| CDVAE | 1.91 | 3.21 |
| DiffCSP | 6.61 | 11.08 |
| AssembleFlow | 2.67 | 3.86 |
| SinkFast | 2.66 | 3.83 |

Table E.6: 1st quantile performance in Å($\downarrow$) of our proposed SinkFast and AssembleFlow rigid-body methods against inorganic crystal structure prediction models CDVAE and DiffCSP on COD_Cluster17 - 5k test set : filtered on the CDVAE subset.

| Method | $\mathbf{PM}^*_{\mathbf{center}}$ | $\mathbf{PM}^*_{\mathbf{atom}}$ |
| --- | --- | --- |
| CDVAE | 2.55 | 4.67 |
| DiffCSP | 9.61 | 15.19 |
| AssembleFlow | 2.77 | 4.43 |
| SinkFast | 2.84 | 4.23 |

**Extension to matching loss.**  A Sinkhorn-based matching loss could also be applied to inorganic CSP models. In this non-rigid-body case the model would not be penalized when identical atoms are correctly predicted but swapped. We have tried to train a simple adaptation of DiffCSP with this matching loss. However, as a diffusion based model, DiffCSP predicts the noise that has been applied to each atomic position to denoise it and progressively reconstruct the crystal. The target is then a noise attached to each atom and the Sinkhorn-based matching loss applicability is more complex and questionable.

Similarly, in order to make it work with flow matching on AssembleFlow, we have had to make some adjustments. For instance, an initial reassignment needs to be computed before interpolating trajectories $X_t$ between $X_0$ initial positions and $X_1$ final ones which are not the dataset's target ones in this case. Also, we have observed that the model needs to predict $X_1$ from intermediate positions $X_t$ rather than predicting some noise in order to make it work.

We believe our quick implementation of DiffCSP is already not well adapted to organic CSP. This leads to serious limitations at the current time for this task and questions the motivation to increase the complexity of it with a Sinkhorn matching loss. However it could be adapted and applied to inorganic CSP datasets such as the Materials Project. We believe it could have great potential on this task if appropriately adapted.

### E.2.2 Using pretrained weigths

Neither CDVAE nor DiffCSP was designed for organic crystal prediction. These methods predict the crystal lattice—central to their original formulation—which is not part of our setting. Nevertheless, we performed another comparison by reusing pretrained versions of these models on inorganic crystals and assuming they transfer reasonably to organic crystals. In this setup, we predict both the lattice and the atomic positions of a single molecule in the unit cell, following the original design of these models. We then reconstruct a cluster of 17 nearest-neighbor molecules under the P1 space group (as these models are not conditioned with a specific space group) and compare this cluster to the ground-truth COD-Cluster17 data. The results are shown in Table 1. Moreover, as DiffCSP is particularly faster than cond-CDVAE we have been able to also evaluate its performance when predicting a number of molecules in the unit cell corresponding to the space group of the target crystal. The results of this experiment are given under the DiffCSP-MultMol line and show a clear improvement.

However, it is essential to note that the limitations of such a comparison to these inorganic models are numerous. In particular, the generated conformations are chemically different from the expected ones. Also, we adapt the setting to predict the unit cell in the P1 space group and then reconstruct 17 nearest neighbors. Finally, the elevated performance of both atom-level models suggests a limitation of current evaluation metrics, which are more suitable for assessing rigid-body-based models. We argue that additional metrics are needed to enable a fairer comparison between rigid-body and atom-level methods. For example, one can assess the validity of the produced chemical graphs and then compare them to the ones from the ground-truth crystallographic molecules.

### E.3 Dependence to the correctness of the conformation

To evaluate our model's dependency on the correctness of the initial molecular conformations, and to support the rigid molecule formulation of the initial packing probelm, we conducted the following experiment. For each molecule in the COD-Cluster17-5k test set, we extracted the corresponding SMILES representation of the ASU molecule and generated five stable conformations using RDKit (Landrum et al., 2025), using EmbedMolecule followed by UFFOptimizeMolecule functions. Each generated conformation is then passed through our model to predict the packed molecular positions.

To assess the quality of RDKit-generated conformtations, we computed the symmetry-corrected RMSD values between RDKit-generated conformation and crystallographic structures using the spyrmsd algorithm (Meli & Biggin, 2020) from RDKit (Landrum et al., 2025) and present the results in Figure E.1. We can see that about 25% of the generated conformations are sufficiently close to the crystallographic ones (within 2Å RMSD) and the median RMSD is below 4Å. This experiment supports the rigid-body approximation in our model. We also computed Packing Matching (PM) between each RDKit sampled molecule conformation and its corresponding COD-Cluster17 conformation. On average, PM was 3.27 Å with a standard deviation of 2.19 Å and a median of 3.11 Å. Due to RDKit failures on 170 of the 500 test set structures caused by issues such as improper valences or atom count mismatches–typically to experimentally invisible hydrogens–our analysis focuses on a subset of 330 molecules, referred to as the RDKit subset.

The results are presented in Tables E.7, E.8 and E.9 under the RDKit column. First, we compare performance on RDKit-generated versus crystallographic conformations for both SinkFast and AssembleFlow. In terms of center-of-mass alignment ($PM_{center}$), the methods perform comparably across the two types of input. However, the performance are slightly hindered in the atom-to-atom comparison. This shows that conformations are not well represented in our model. Second, comparing Table E.8 to Table E.9 we observe that both methods perform much better on crystallographic structures from which we generate five different conformations that are close to the crystallographic ones. This confirms the importance of initial conformational accuracy. However, we suspect a correlation between the size of the rigid molecule and how close are conformations generated by RDKit. The good performance of the model could also be explained through this aspect.

Our conclusion is that while the models get a sense of how important initial conformations are, the learned representations are independent to the molecular conformations. We therefore believe that future models should be trained end-to-end, jointly learning conformation and crystal structure prediction. This represents

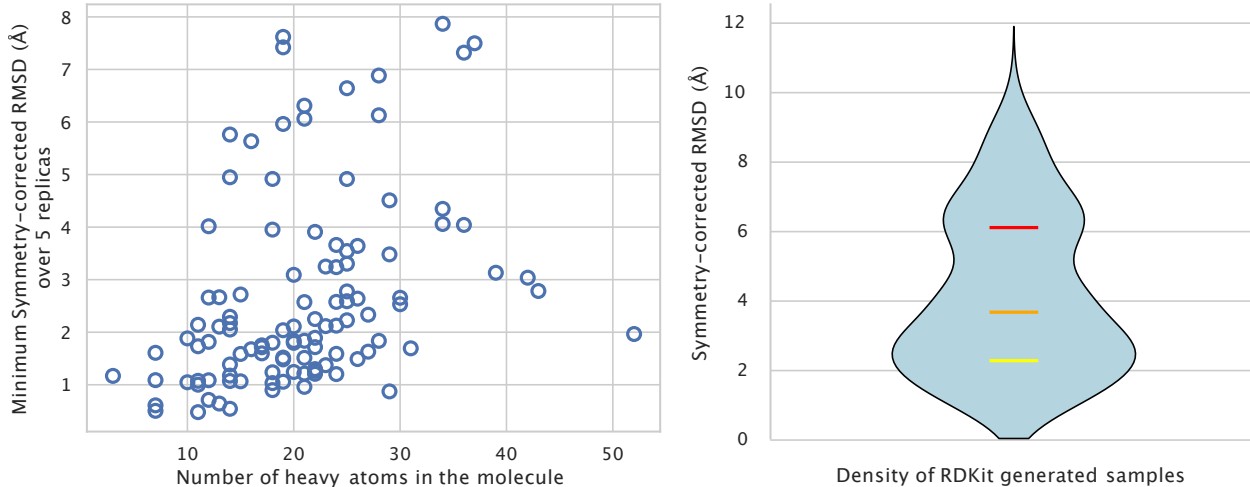

Figure E.1: Left: Distribution of minimum symmetry-corrected RMSD values (Å) over 5 RDKit conformations with respect to the number of heavy atoms in the ASU molecule. Symmetry-corrected RMSD values were computed between RDKit-generated conformations and crystallgoraphic structures with spyrmsd (Meli & Biggin, 2020). Right: Distribution of symmetry-corrected RMSD values between RDKit-generated conformations and crystallgoraphic structures. The yellow bar indicates the first quartile, the orange one the median and the red one the last quartile.

a promising direction for advancing research in this very complex domain. We believe our study helps to identify key challenges and can serve as a foundation for future work in organic crystal structure prediction.

Table E.7: Performance in Å($\downarrow$) of our proposed SinkFast and AssembleFlow methods on both crystallographic and RDKit generated conformations on COD-Cluster17-5k test set : filtered on the RDKit subset.

| Method | RDKit | $\mathbf{PM}^*_{\mathbf{center}}$ | $\mathbf{PM}^*_{\mathbf{atom}}$ |
|---|---|---|---|
| AssembleFlow | | $3.54_{\pm0.01}$ | $5.44_{\pm0.00}$ |
| AssembleFlow | $\checkmark$ | $3.58_{\pm0.00}$ | $5.59_{\pm0.08}$ |
| SinkFast | | $3.59_{\pm0.13}$ | $5.41_{\pm0.08}$ |
| SinkFast | $\checkmark$ | $3.55_{\pm0.13}$ | $5.53_{\pm0.15}$ |

Table E.8: Performance in Å($\downarrow$) of our proposed SinkFast and AssembleFlow methods on both crystallographic and RDKit generated conformations on COD-Cluster17-5k test set : filtered on the RDKit subset with the **lowest packing matching distance** to original ones.

| Method | RDKit | $\mathbf{PM}^*_{\mathbf{center}}$ | $\mathbf{PM}^*_{\mathbf{atom}}$ |
|---|---|---|---|
| AssembleFlow | | $3.27_{\pm0.01}$ | $4.92_{\pm0.03}$ |
| AssembleFlow | $\checkmark$ | $3.27_{\pm0.03}$ | $4.90_{\pm0.03}$ |
| SinkFast | | $3.28_{\pm0.13}$ | $4.88_{\pm0.11}$ |
| SinkFast | $\checkmark$ | $3.18_{\pm0.11}$ | $4.81_{\pm0.12}$ |

## F    Visualizations

Figure F.1 shows the packing of three assemblies randomly picked from the test set. We visualize all atoms as van der Waals (vdW) spheres. We took the standard vdW radii for chemical elements, colored using JMol colors and ray-traced the scenes with PyMol. The image does not demonstrate common patterns, only certain *packing* similarities. One can conclude on the generally poor reconstruction obtained from the two compared

Table E.9: Performance in Å($\downarrow$) of our proposed SinkFast and AssembleFlow methods on both crystallographic and RDKit generated conformations on COD-Cluster17-5k test set : filtered on the RDKit subset with the **highest packing matching distance** to original ones.

| Method | RDKit | $\text{PM}^*_{\text{center}}$ | $\text{PM}^*_{\text{atom}}$ |
|---|---|---|---|
| AssembleFlow | | $3.80_{\pm 0.03}$ | $5.95_{\pm 0.08}$ |
| AssembleFlow | ✓ | $3.89_{\pm 0.01}$ | $6.27_{\pm 0.09}$ |
| SinkFast | | $3.88_{\pm 0.11}$ | $5.92_{\pm 0.00}$ |
| SinkFast | ✓ | $3.92_{\pm 0.12}$ | $6.25_{\pm 0.14}$ |

algorithms. Indeed, the method and the problem formulation do not allow to generalize well enough to be applied and used at large scale.

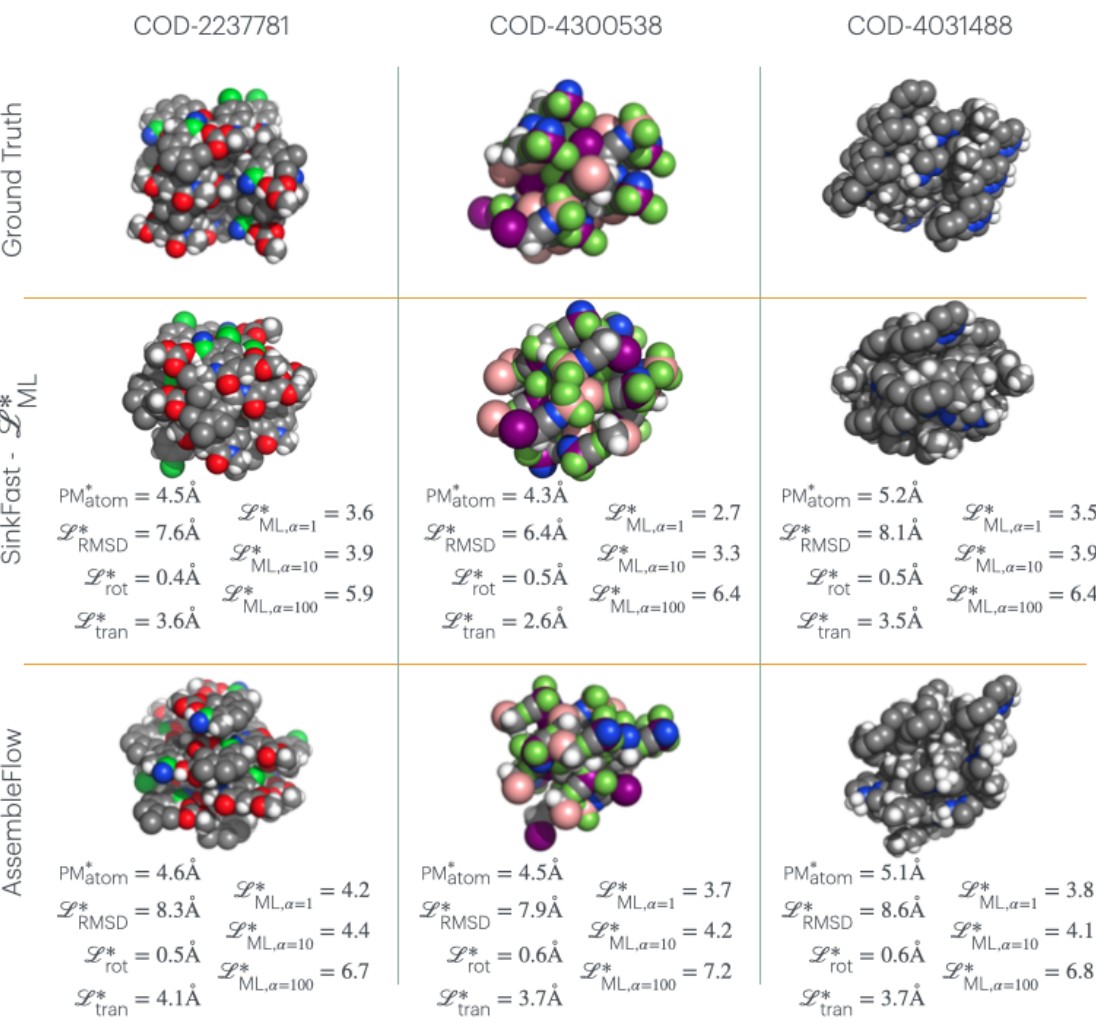

Figure F.1: Visualization of our SinkFast-$\mathcal{L}^*_{\text{ML}}$ prediction against ground truth and AssembleFlow method on 3 examples randomly picked from the test set. Scores of each prediction are reported with $\text{PM}^*_{\text{atom}}$, $\mathcal{L}^*_{\text{RMSD}}$, $\mathcal{L}^*_{\text{tran}}$ the translational error, $\mathcal{L}^*_{\text{rot}}$ the rotational error and 3 $\mathcal{L}^*_{\text{ML}}$ errors with different values of the $\alpha$ parameter. Atoms are colored using the JMol color convention and shown using PyMol molecular visualization system (Schrödinger, LLC, 2015).

