# OpenReview forum: "Challenges in Non-Polymeric Crystal Structure Prediction: Why a Geometric, Permutation-Invariant Loss is Needed"
_TMLR — Accepted by TMLR_

### Review · Reviewer_NgFZ · 2026-02-01

**Summary Of Contributions:**

The problem definition and challenges are still not clear to me.
Could the authors explain:

Why an infinite structure is ‘ defined by a
finite number of symmetry operators applied to the ASU?’

In fig 1. it seems the thing is to compile ASUs up following some pattern. Then isn’t the thing about predicting several rotation and translation matrices, which can then be applied repeatedly resulting in the infinite structure?

Despite the concern about whether the task is defined appropriately, the results demonstrate superiority. And the result that regression beats diffusion / flow matching shows good efficiency and effectiveness.

**Audience:**

Yes

**Audience Explanation:**

This is a practical scientific task. People in the relevant areas should be interested.

**Claims And Evidence:**

Yes

**Claims Explanation:**

I chose yes, but I am not so sure. The experimental results show that the proposed method beats baseline.
But I am also not sure if I understand the problem.

**Requested Changes:**

I wonder if figure 1 and the textual content correctly define the task and motivate this work. In my understanding, the task is: given a set of identical rigid molecules randomly placed in space, predict how they pack together to form a crystal-like structure. If this is the case, since the molecules are all identical, and according to fig 1, the arrangement follows some rule, why don’t we just predict a set of translations and rotations based on ONE molecule? For example, we have 17 identical molecules, why not use one as input and ask the model to output 17 translations and rotations?

---

### Review · Reviewer_czGH · 2026-02-03

**Summary Of Contributions:**

This paper addresses the molecular assembly problem, a subtask of non-polymeric crystal structure prediction, where a set of identical rigid molecules is packed into a crystalline structure. The authors argue that existing methods use ill-posed training objectives and propose improved loss functions to address this issue by permutation invariant method.

### Main Contributions

**1. Physics-grounded Rigid-body Loss ($\mathcal{L}_{RMSD}$)**

Previous methods decouple translation ($\mathbb{R}^3$) and rotation ($SO(3)$), requiring a hyperparameter α to balance them. The authors propose a RMSD-based loss utilizing the inertia tensor, which naturally integrates both components without additional hyperparameters and provides physically meaningful rigid-body comparison.

**2. Permutation-invariant Loss with Sinkhorn Algorithm**

The authors point out that identical molecules within a unit cell are physically equivalent, making the enforced index correspondence in existing losses inappropriate. They introduce differentiable optimal assignment via the Sinkhorn algorithm to achieve permutation invariance during training.

**3. Evaluation Metric (PM\*)**

The paper proposes PM\*, a permutation-invariant version of the existing Packing Matching (PM) metric, enabling more physically appropriate evaluation of molecular assembly predictions.

### Key Strengths

- Achieves state-of-the-art performance with simple regression, resulting in faster training and faster inference.
- The loss function design has physical justification based on rigid-body mechanics and crystallographic equivalence of identical molecules.
- Ablations on flow matching, permutation invariance, and loss function variants provide clear insights into each component's contribution.

### Key Weaknesses

- Experiments are conducted on COD-Cluster17, leaving generalization to other datasets (e.g., OMC25) unexplored.
- Some baselines (DiffCSP, CDVAE) are evaluated under a different setup from their original design, which may affect fair comparison.
- Energy-based evaluation using pretrained MLIPs or property prediction is not included, despite the paper's motivation about "materials with targeted properties."
- Comparisons with more recent methods (e.g., FlowMM, OMatG) are not provided.

**Audience:**

Yes

**Audience Explanation:**

The paper addresses a relevant problem in computational materials science, specifically the molecular assembly task for crystal structure prediction. The proposed permutation-invariant loss design could be of interest to researchers working on molecular crystal modeling with geometric deep learning.

**Broader Impact Concerns:**

I have no concerns regarding the ethical implications or broader social impact of this work.

**Claims And Evidence:**

Yes

**Claims Explanation:**

The main claims of this paper are well-supported by experimental evidence:

1. **Effectiveness of the proposed loss functions**: The ablation study in Table 2 compares different loss variants ($L_{ML}, L_{RMSD}, L_{Geom}$) with and without permutation invariance, demonstrating the contribution of each component. The results show consistent improvements when using permutation-invariant losses across both the 5k subset and the full dataset.

2. **Performance gains over baselines**: Table 1 shows that SinkFast outperforms all baselines on both standard PM metrics and the proposed PM* metrics, providing evidence that the improvements are not solely due to the new evaluation metric.

3. **Computational efficiency**: Table 3 provides concrete timing comparisons, showing 42× faster training and 50× faster inference compared to flow matching approaches, which supports the claim that simple regression with proper loss design can replace more complex generative frameworks.

**Requested Changes:**

1. **Additional dataset evaluation**: Evaluating on at least one additional dataset (e.g., OMC25 or a subset of it) would strengthen the generalization claims. This would help demonstrate that the proposed loss functions are not overfitted to the specific characteristics of COD-Cluster17.

2. **Energy-based validation**: Including energy evaluation of predicted structures using a pretrained MLIP (e.g., UMA or MACE) would provide complementary evidence that the predicted structures are not only geometrically accurate but also physically plausible.

3. **Clarification on baseline comparisons**: Some baselines (DiffCSP, CDVAE) predict all-atom coordinates, while SinkFast predicts molecule-level rotations and translations. A more detailed discussion on how this fundamental difference affects the comparison would improve transparency.

4. **Ablation on flow matching**: An experiment comparing AssembleFlow with the proposed L*_RMSD loss in a one-step setting would help isolate the contribution of the loss function from the choice of regression over flow matching.

---

> ### Comment · Reviewer_czGH · 2026-03-27
>
> Thank you for the thorough revision and additional experiments. The new experimental results and the extended baseline comparisons are appreciated and help to strengthen the paper.

---

### Review · Reviewer_bCnM · 2026-02-19

**Summary Of Contributions:**

This paper argues that the previous learning objective used in the COD-Cluster17 molecular assembly benchmark is ill-posed because it relies on an arbitrary, index-based correspondence between identical molecules. To address this issue, the authors reformulate training around rigid-body poses in SE(3) and introduce (i) a physically grounded rigid-body loss based on RMSD with inertia tensor weighting, and (ii) a permutation-invariant training objective implemented via differentiable soft assignment using the Sinkhorn algorithm.

Using the same backbone architecture as prior work, the authors show that a regression model trained with the proposed permutation-invariant losses matches or improves state-of-the-art packing matching scores on COD-Cluster17 (both the 5k subset and full dataset), while substantially reducing training cost compared to flow matching approaches. The paper further provides ablations comparing losses and training schemes, and discusses benchmark limitations and remaining challenges, particularly poor orientation accuracy.

## Strengths

1. The paper clearly identifies the conceptual mismatch between physical equivalence of molecules and index dependent supervision and evaluation in the benchmark.
2. The learning objective design is conceptually sound, combining permutation invariance with a physically motivated rigid-body loss and a relational geometric objective.
3.  The method demonstrates consistent improvements under controlled comparisons on COD-Cluster17, alongside significant training time reductions compared to flow matching.

## Weaknesses

1. Results are confined to a single benchmark with known physical limitations, including the absence of periodic boundary conditions and space-group information.
2. Molecular orientations are poorly predicted, with limited investigation of the underlying causes.

**Audience:**

Yes

**Audience Explanation:**

The paper addresses questions related to permutation ambiguity, symmetry considerations, and learning objective design, which are broadly relevant themes in machine learning. However, the concrete problem setting and empirical validation are centered on a highly specific molecular assembly benchmark with known structural limitations. Consequently, the practical implications of the results may be most directly relevant to researchers working on geometric or scientific machine learning problems closely aligned with this setting. Nevertheless, the conceptual discussion on permutation invariance and objective design may still be of interest to a subset of the TMLR audience.

**Broader Impact Concerns:**

No significant broader impact or ethical concerns.

**Claims And Evidence:**

Yes

**Claims Explanation:**

The main empirical claims are supported by experimental evidence. The paper provides controlled comparisons using the same backbone architecture as prior work, which isolates the effect of the proposed learning objectives. The reported improvements on COD-Cluster17 are consistent across both the 5k subset and the full dataset, and are further supported by ablation studies comparing alternative losses and training schemes.

In particular, the analysis highlighting the impact of permutation invariance on evaluation metrics is clear and convincing. The computational efficiency claims are also substantiated through explicit training time comparisons. While some broader interpretations (e.g., regarding the role of flow matching) could benefit from deeper analysis, the core claims of the paper are adequately supported by the presented results.

**Requested Changes:**

While the paper identifies a meaningful issue in the problem formulation of prior non-polymeric CSP studies and proposes a well motivated permutation invariant, physics based loss, I have concerns regarding the scope and physical relevance of the experimental setting.

1. The paper’s conclusions are closely tied to the COD-Cluster17 benchmark, whose formulation removes periodic boundary conditions (PBC), thereby weakening the connection to realistic CSP settings. Although the proposed physics based, permutation invariant loss remains conceptually valid under periodic formulations, its relative necessity may be substantially reduced when PBC constraints are present. In periodic unit cell descriptions, the space of admissible molecular assignments is inherently more restricted than in supercell derived cluster systems, leading to significantly fewer optimal transport ambiguities.

More generally, it remains unclear whether some of the observed difficulties, particularly the poor recovery of molecular orientations, are intrinsic to the proposed learning objectives or instead consequences of the non periodic cluster formulation. Predicting structures directly under periodic unit cell constraints might naturally encode physically meaningful restrictions, potentially reducing the need for explicitly physics grounded loss constructions. If accurate periodic or unit cell structures cannot be reliably reconstructed from predictions obtained in a PBC free cluster setting, the suitability of such benchmarks as primary evaluation testbeds also becomes less clear. Clarifying these distinctions would strengthen the paper.

2. The restriction of experiments to COD-Cluster17 further limits the strength of the claims. The decision not to evaluate on larger or more structurally diverse datasets such as OMC is not entirely convincing. Although simulated datasets differ from experimentally resolved structures, they provide controlled and scalable environments that are well suited for testing invariance properties and learning objectives. Demonstrating that the proposed scheme remains effective under more complex or varied conditions would substantially strengthen the work.

---

### Decision · Action_Editor_m1p4 · 2026-04-01

**Recommendation:** Accept as is

**Audience:**

Yes

**Audience Explanation:**

The paper addresses permutation ambiguity in structured prediction, which is a broadly relevant theme in geometric and scientific ML. The specific finding that correcting the training objective alone allows simple regression to match or outperform sophisticated generative models (flow matching) carries a useful methodological lesson beyond the molecular assembly setting. Researchers working on crystal structure prediction, symmetry-aware learning, and loss design for scientific applications would benefit from these findings.

**Claims And Evidence:**

Yes

**Claims Explanation:**

The ablation studies isolate the effect of:
- permutation invariance
- physically grounded rigid-body loss across multiple loss variants
- training schemes (regression vs. flow matching)

The proposed method is validated on two datasets (COD-Cluster17 and OMC25-Cluster17 added during rebuttal).

The authors extended baseline comparisons with pretrained DiffCSP/CDVAE and a new single-molecule baseline suggested by Reviewer NgFZ.